# T-cell dysfunction in the glioblastoma microenvironment is mediated by myeloid cells releasing interleukin-10

Vidhya M. Ravi [1,2,3,4,5,19], Nicolas Neidert [1,2,3,6,19], Paulina Will [1,2,3,19], Kevin Joseph [1,2,3], Julian P. Maier [1,2,3], Jan Kückelhaus[1,2,3], Lea Vollmer[1,2,3], Jonathan M. Goeldner [1,2,3], Simon P. Behringer[1,2,3,6], Florian Scherer [3,7], Melanie Boerries[3,8,9,10], Marie Follo[3,7], Tobias Weiss [11], Daniel Delev[12,13], Julius Kernbach [12,13], Pamela Franco[2,3,6], Nils Schallner[3,14], Christine Dierks[3,7], Maria Stella Carro [2,3], Ulrich G. Hofmann[2,3,4], Christian Fung[2,3], Roman Sankowski [3,15], Marco Prinz [3,15,16,17], Jürgen Beck[2,3,17], Henrike Salié [3,16,18], Bertram Bengsch [3,16,18], Oliver Schnell[2,3,6,19] & Dieter Henrik Heiland [1,2,3,9,10,19 ✉]

Despite recent advances in cancer immunotherapy, certain tumor types, such as Glioblastomas, are highly resistant due to their tumor microenvironment disabling the anti-tumor immune response. Here we show, by applying an in-silico multidimensional model integrating spatially resolved and single-cell gene expression data of 45,615 immune cells from 12 tumor samples, that a subset of Interleukin-10-releasing *HMOX1*+ myeloid cells, spatially localizing to mesenchymal-like tumor regions, drive T-cell exhaustion and thus contribute to the immunosuppressive tumor microenvironment. These findings are validated using a human ex-vivo neocortical glioblastoma model inoculated with patient derived peripheral T-cells to simulate the immune compartment. This model recapitulates the dysfunctional transformation of tumor infiltrating T-cells. Inhibition of the JAK/STAT pathway rescues T-cell functionality both in our model and in-vivo, providing further evidence of IL-10 release being an important driving force of tumor immune escape. Our results thus show that integrative modelling of single cell and spatial transcriptomics data is a valuable tool to interrogate the tumor immune microenvironment and might contribute to the development of successful immunotherapies.

A full list of author affiliations appears at the end of the paper.

Tumor-infiltrating lymphocytes, along with resident and infiltrating myeloid cells, make up a significant proportion of the glioblastoma tumor microenvironment[1–3]. Recently, single-cell RNA sequencing (scRNA-seq) based characterization of the myeloid cell compartment revealed remarkable heterogeneity with regards to cellular diversity and transcriptional plasticity[1,4]. However, the variety of lymphoid cell types within malignant brain tumors remains poorly investigated. Insights into the transcriptional programs and genetic drivers of lineage differentiation within the lymphoid compartment will aid in understanding key mechanisms that play important roles in the T cell differentiation, within the glioblastoma microenvironment. Recently, two studies have provided evidence that the crosstalk between myeloid and lymphoid cells accounts for a significant degree of T cell malfunction, which partially explains the lack of antitumor immunity[5,6]. In other cancer entities such as colorectal cancer[7], liver cancer[8], or melanoma[9], a variety of T cell states have been identified and investigated. Prolonged immune activation and ambiguous stimulation, such as those seen during uncontrolled tumor growth or chronic infection, alter the CD8[+] lymphocyte secretome, resulting in a loss of their cytotoxic profile[9–11], also referred to as T cell exhaustion. This exhaustion state is marked by inhibitory cell surface receptors (PD1, CTLA-4, LAG-3, TIM3, and others), in addition to anti-inflammatory cytokines such as IL-10 and TGF-ß present in the tumor microenvironment[11–13]. Therefore, this cellular state of "dysfunction" or "exhaustion" represents a paramount barrier in the development of successful immune-based vaccines or checkpoint therapy[2,14,15]. Glioblastoma, the most common and aggressive primary brain tumor in adults, is archetypical for tumors with a strong immunosuppressive microenviroment[16]. Current, immunotherapies such as PDL1/PD1 checkpoint blockade[17] or peptide vaccination[18], that have led to remarkable improvement in therapeutic outcome for several types of cancer, has failed to demonstrate its effectiveness in patients suffering from glioblastoma.

To address this sparsity of knowledge with respect to the lymphoid cell population in glioblastoma, we performed transcriptional profiling using scRNA-sequencing, mapping potential cellular interactions and/or cytokine responses that could lead to dysfunctional and/or exhausted T cells. Pseudotime analysis revealed an increased response to Interleukin 10 (IL-10) during the transformation of T cells from the effector state to the dysfunctional state. To computationally explore the "connected" cells driving this transformation, we introduce an in silico approach termed "Nearest Functionally Connected Neighbor (NFCN)", which identified a subset of myeloid cells, marked by *CD163* and *HMOX1* expression. Further, spatially resolved transcriptomics confirmed the spatial overlap between exhausted T cells and *HMOX1*[+] myeloid cells, spatially localized within regions of the tumor enriched for mesenchymal transcriptional signatures. Finally, using a human neocortical GBM model coupled with patient-derived T cells to simulate the lymphoid compartment, we validated the role of HMOX1[+] myeloid cells as key drivers for the immunosuppressive microenvironment found in glioblastoma. The dysfunctional T cell transformation was found to be rescued by the inhibition of the JAK-STAT pathway, mediated by a reduction in IL-10 release in our ex-vivo model, as previously demonstrated[19]. Based on these results, we treated a single recurrent glioblastoma patient in a neoadjuvant setting, with JAK-STAT inhibitor (Ruxolitinib), partially rescuing the immunosuppressive environment.

Here, we show the interaction between the organ-specific structural immunity of the brain and infiltrating lymphoid cells in glioblastoma illustrating an exhausted T cell state in close proximity to mesenchymal differentiated tumor cells. We demonstrate that a defined myeloid cell population contributes to T cell dysfunction through IL-10 signaling. In vitro and vivo data reveal that manipulation of the driver pathway can rescue the immunosuppressive impact of the myeloid cell population. Our findings open perspectives to target the tumor microenvironment and to improve immunotherapy response in glioblastoma.

## Results

**Single-cell analysis of the immune cell compartment in glioblastoma.** In order to decipher the diversity of the immune compartment within the glioblastoma microenvironment, we performed droplet-based single-cell sequencing of neoplastic tissue samples from 8 patients, diagnosed with de novo glioblastoma (Detailed clinical data available in Supplementary data 1). Lymphoid and myeloid populations (CD45[+]/CD3[+]) were sorted from neoplastic tissue specimens (Fig. 1a and Supplementary Fig. 1). The scRNA-seq dataset consisted of 45,615 cells, with a mean of 9956 unique molecular identifiers (UMIs) and ~2989 uniquely expressed genes per cell, Fig. 1b and Supplementary Fig. 2. The data were corrected for mitochondrial gene expression, cell cycle effects were regressed out, and batch effects due to technical artifacts were accounted for. Horizontal data integration was performed using the mutual nearest neighbor algorithm (MNN), using the top 2000 most variable features as anchors, Supplementary Fig. 3. Cell types were inferred using weighted mutual neighbor (WNN)[20] vertical integration with a publicly available reference dataset[20]. To ensure comparability with previously published datasets, we performed cross-validation with reference dataset, demonstrating a close distribution of T cell subtypes within all datasets[21,22], Supplementary Fig. 2d. To identify and exclude malignant cells, we inferred large-scale copy-number variations (CNVs) from scRNA-seq profiles by averaging expression over stretches of 100 genes on their respective chromosomes[23]. With this approach, we confirmed that there was minimal contamination by tumor cells (clustered as OPC cells), based on their typical chromosomal alterations (gain of chromosome 7 and loss of chromosome 10), Supplementary Fig. 4.

**Diversity of T cell states in the glioblastoma microenvironment.** To explore the diversity of the T cells that are present in the glioblastoma microenvironment, we isolated CD4[+] and CD8[+] T cells by two different but complementary in silico methods. T cells were first identified by means of non-negative matrix factorization (NMF) and shared-nearest neighbor (SNN) clustering. Cell type assignment was carried out based on marker gene profiles (CD3[+], CD4[+]/CD8[+]). In the second method, T cells were isolated using a WNN-classification model, which resulted in a total of 7352 cells (3602 CD4[+] cells and 3750 CD8[+] cells) Fig. 1b, c. For CD4[+] and CD8[+] T cells, NMF-based clustering was performed, with the optimal cluster resolution determined by its cluster-tree stability which resulted in six CD8[+] and CD4[+] clusters, Fig. 1c.

Clusters from the CD8[+] T cell populations spanned several T cell states, including the effector T cell program (**CD8[+]Teff:** *EOMES, GZMY, GZMK,* and *CTRAM*), effector memory program (**CD8[+]TEM:** *IER2* and *BCL2*), tissue-resident memory programs (**CD8[+]TRM-CD39:** *ITGAE* and *ZNF6839*), exhausted T cells (**CD8[+]Texhaused:** *HAVCR2, PDCDL1, CTLA4,* and *CXCL8*), heat shock protein-expressing or stress associated (**CD8[+]HSP** *HSPA1A*) and a transcriptional program similar to a mucosal-associated invariant T cell (**CD8[+]MAIT-like** *CCR6, CD226,* and *RORA*), Fig. 1d.

In the CD4[+] cell population, we identified cells enriched for effector memory programs (**CD4[+] TEM:** *CD44, TAF4B,* and *RASA3*), two T helper programs (**CD4[+]Th1:** *TBX21, IFNG,* and

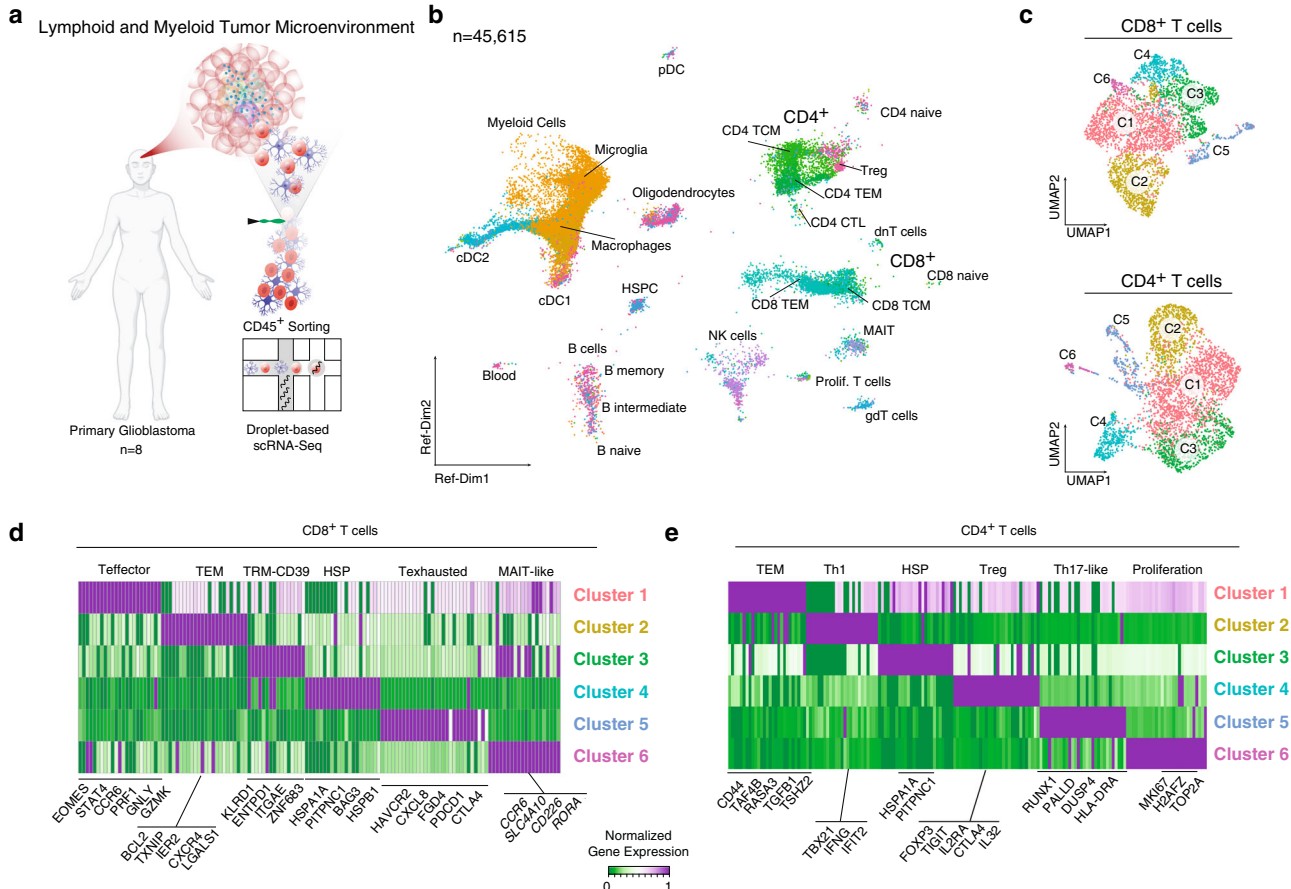

**Fig. 1 Cluster and cell type analysis of lymphoid cells in glioblastomas. a** Illustration of the workflow: tissue specimens for scRNA-sequencing were obtained from eight donors diagnosed with primary glioblastoma. **b** Weighted mutual neighbor (WNN) data integration and dimensional reduction using reference UMAP, was used to determine the cell types. **c** Isolated and clustered CD8+ (upper UMAP) and CD4+ (bottom UMAP) cells are indicated by colors. **d, e** Heatmaps of marker genes (top 30) across clusters from CD8+ cells (right panel) and CD4+ cells (left panel). T effector T effector cells, TEM T effector memory, TRM tissue-resident memory cells, TCL T cell lymphoma, TCM central memory, gdT gamma-delta T cells, MAIT mucosa-associated T cells, dnT double-negative T cells, Treg regulatory T cells, pDC progenitor dendritic cells, cDC conventional dendritic cell, NK natural killer cells, Th1 T helper cells, HSP heat shock protein cell type, CTL cytotoxic T-cells, HSPC hematopoietic stem and progenitor cell.

*IFIT2*; **CD4⁺Th17-like** *RUNX1, PALLD, DUSP4, HLA-DRA*, similar to Th17/ invariant Treg signatures[24]), a stress-associated cluster (**CD4⁺HSP** *HSPA1A*), a cluster with regulatory T cell program (*FOXP3, TIGIT,* and *IL2RA*) and a cluster of proliferating T cells (*MIK67* and *TOP2A*), Fig. 1e. Due to the absence of proliferating CD8⁺ and only a minimal proportion of proliferating CD4⁺ T cells, our data support the general observation that antitumor immunity in GBM is weak. To investigate this putative tumor-associated dysfunction, we estimated the likelihood of T-cell dysfunction/exhaustion based on known signature gene expression within each cluster. CD8⁺ T cells revealed a stronger probability for tumor-associated exhaustion compared to the relatively exhaustion-resistant CD4⁺ population[25]. In CD8⁺ T cell populations, the strongest association to tumor-associated exhaustion was confirmed in the clusters 5 and 3 (TRM-CD39 and T exhausted) Fig. 2a, b (FDR < 0.01, hypergeometric test).

**Tumor-associated T cell exhaustion correlates with IL-10 response**. To identify potential drivers of T cell exhaustion, hierarchical trees were reconstructed to infer cellular differentiation trajectories. This was then re-embedded into a dynamic model (RNA-velocity) to decipher the temporal evolution of T cell states. We used a single-cell trajectories reconstruction, exploration, and mapping (STREAM) model, to identify lineage states in both CD8⁺ and CD4⁺ T cell populations. For the CD8⁺

population, we identified three major branches (S0–S2), reflecting major T cell states, Fig. 2b. The "S1" state was found to be associated with increased expression of T cell exhaustion/dysfunction markers (*HAVCR2, CTLA4,* and *PDCD1*) and enriched cells from the T exhausted cluster. The "S2" state predominantly contained effector T cells, Fig. 2c. When we mapped the gene expression of previously defined exhaustion and effector markers along the pseudotime trajectory spanning from state S0 to state S1, an enrichment of the exhaustion gene expression program was observed at the terminus of the S1 branch, Fig. 2d. In order to determine transcriptional pathways associated with T cell exhaustion, pathway signaling inference was carried out, revealing a significant correlation between T cell exhaustion and IL-10 as well as partially TGFß response (false discovery rate [FDR] <0.01, hypergeometric test). This finding confirms results presented in previous reports linking IL-10-signaling to tumor-associated T cell exhaustion through the STAT3-BLIMP-1 (gene: *PRDM1*) axis[12]. In the CD4⁺ T cell population, only the relatively small Th17-like cluster showed strong enrichment for T cell exhaustion genes, Fig. 2c. Pseudotemporal lineage reconstruction revealed that the Th17-like cluster has a distinct sub-lineage (CD4⁺ T cell state S3 and S4), marked by increased enrichment of IL10 and TGF-beta response pathways, Fig. 2e, f. Other identified lineages correspond to known CD4⁺ lineages such as regulatory or memory T cells, Fig. 2e.

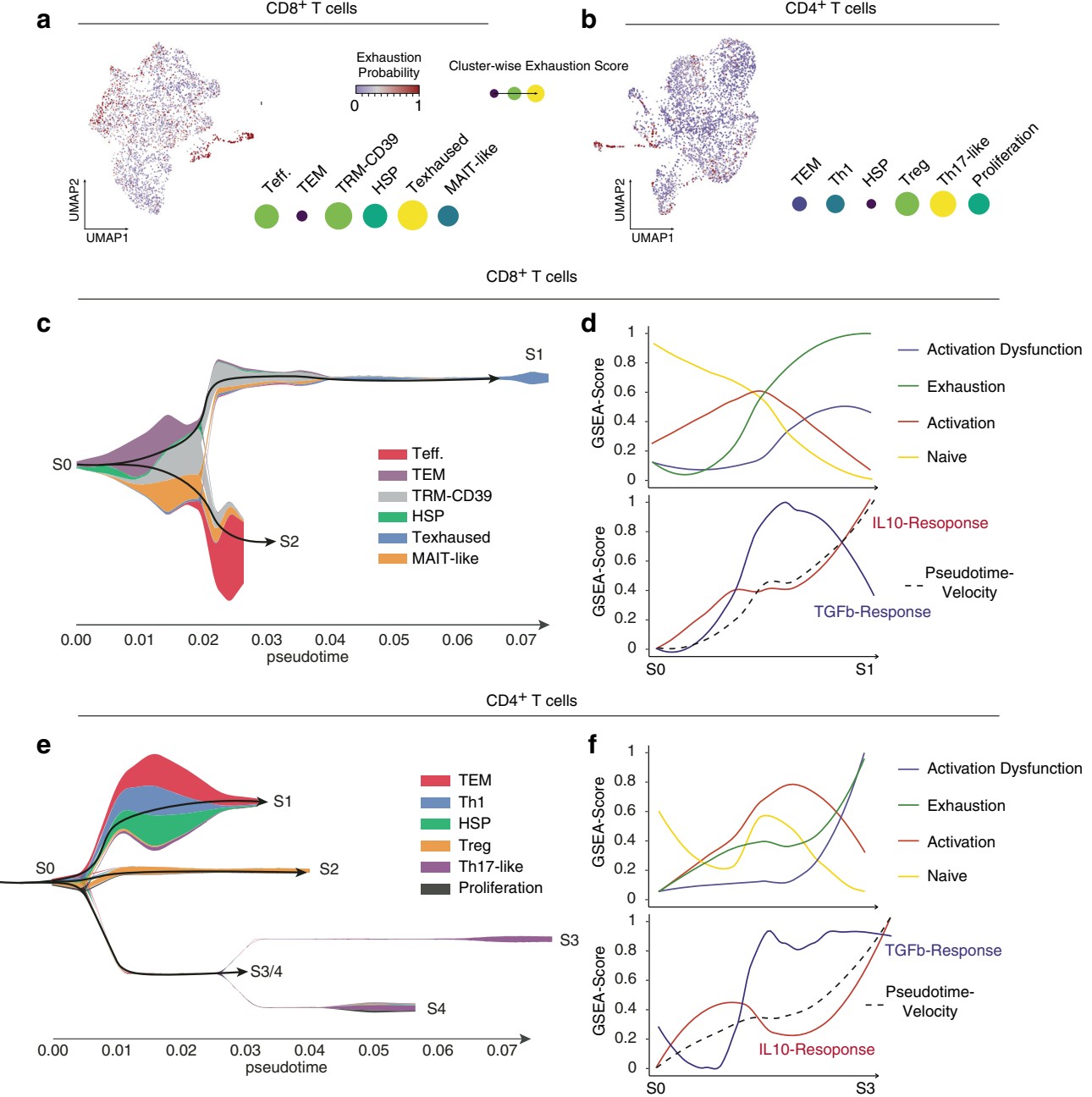

**Fig. 2 Pseudotemporal analysis of T cell differentiation. a, b** UMAP dimensional reduction for CD8+ (left) and CD4+ T cells. Colors indicate the enrichment of T cell exhaustion. Cluster-wise enrichment analysis is illustrated by points, which are sized and colored according to their enrichment score for T cell exhaustion. **c** For CD8+ T cells: The two identified terminal states (indicated by colors, names S1-S2) are presented using a two-dimensional subway plot. Terminally exhausted T cells fall under state 1 (S1). **d** For CD8+ T cells: Line plots with pseudotime (from S0 to S1 state) at the x-axis and enrichment scores of different T cell states on the y-axis (upper plot). At the bottom, the RNA-velocity from S0 to S1 state is illustrated (dashed line) with increasing velocity-pseudotime on the y-axis. Additionally, IL-10 and TGFb response is shown (red and blue lines). **e** Similar visualization (**c**) from CD4+ T cells. These cells reveal a more complex architecture mainly spanning the three major cell states from regulatory T cells (branch S2), TEM/Stress (Branch S1), and Th17 (Branch S3/S4). **f** Similar representation (**d**) for CD4+ T cells with enrichment for T cell states (upper plot) and velocity and IL-10/ TGFb response (bottom plot).

**IL-10 mediated downstream signaling in T cells**. To gain further insights into accurate downstream signaling due to the immunosuppressive cytokine IL-10, we carried out IL-10 stimulation of both naive and pre-stimulated T cells (IL2/IFN-gamma) followed by bulk RNA-seq. IL-2 was used to simulate the early stages of immune response in T cells[26]. De novo motif enrichment was then carried out on genes that were significantly upregulated to infer common transcription factor binding sites, Fig. 3a. In line with literature[12], we confirmed a significant enrichment of *IRF1* and *PRDM1/14* binding sites. A close overlap between both BLIMP-1 (Gene: *PRDM1*) and IRF1 peaks in ChiP-seq data was reported recently, confirming that IL-10 stimulation can be linked to BLIMP-1 transcriptional regulation[27]. Exploration of CD8+ T cells from our scRNA-seq data supported this finding, with increased expression of *PRDM1* in the exhausted CD8+ cluster, Fig. 3b.

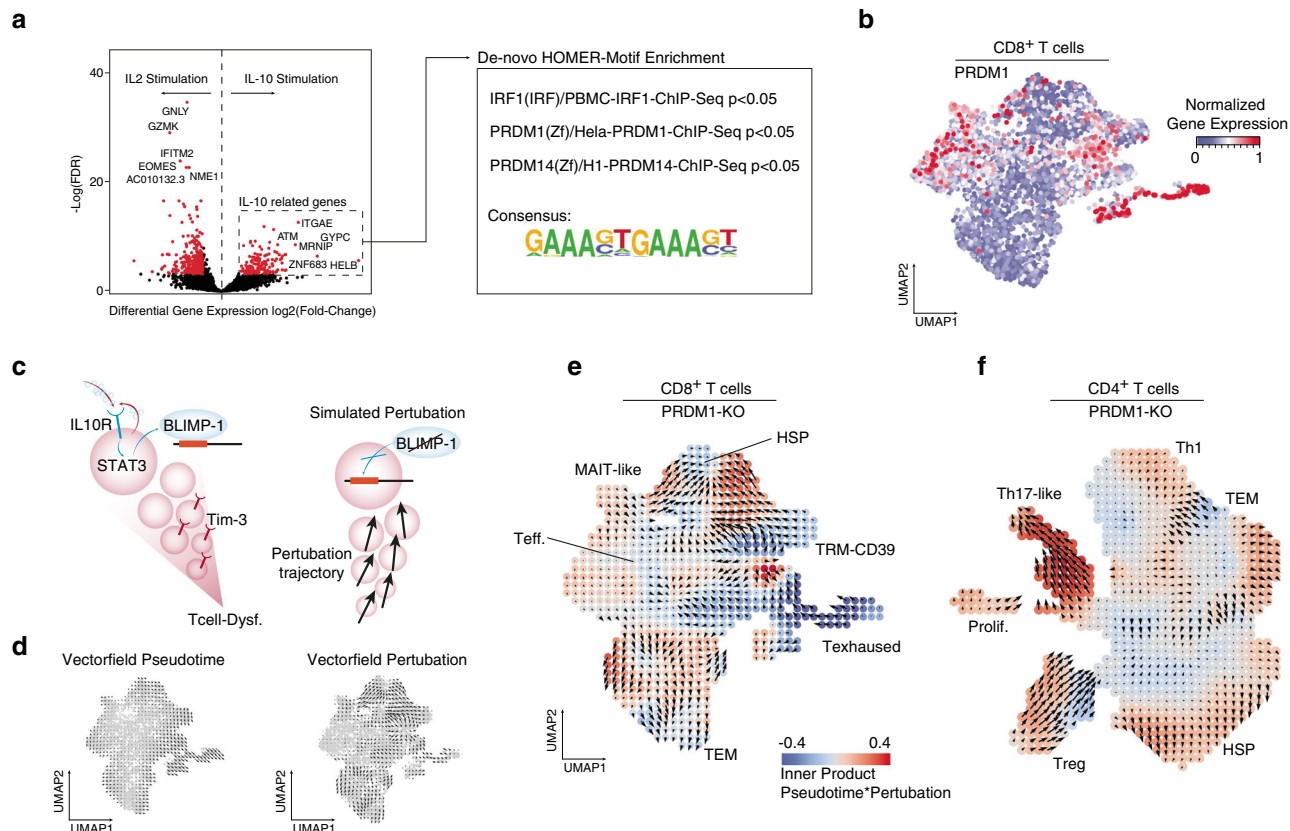

**Fig. 3 Perturbation simulation of IL-10 response in T cells. a** Volcano plot the differential gene expression of IL-2 and IL-10 stimulated T cells. Significantly expressed genes (IL-10 related genes) are analyzed by HOMER (Software for motif discovery and next-generation sequencing analysis) for de novo motif enrichments. The top three candidates are shown in the right box. **b** UMAP representation of CD8+ T cells expressing the PRDM1 gene. **c** Illustration of the hypothesis of PRDM1 perturbation. **d** Vector field projection of the development trajectories (RNA-velocity) (left) and the perturbation trajectories (right) in CD8+ T cells. **e** CD8+ T cells: Vector field projection of the inferred perturbation (PRDM1), colors indicate the inner product from both development and perturbation trajectories. **f** Similar presentation as (**e**) from CD4+ T cells.

Based on the findings so far, we hypothesized that the IL10-STAT3-BLIMP-1 signaling axis could be a potential driver of tumor-associated T cell exhaustion. To further investigate the transcriptional effects of *PRDM1*, we performed in silico perturbation of *PRDM1* within our scRNA-seq dataset. This in silico perturbation required gene regulatory networks (GRNs), inferred using CellOracle[28], Fig. 3c. Integration of pseudotime and perturbation trajectories (Fig. 3c) by its inner product revealed a reversal of differentiation towards T effector programs in CD8+ T cells after PRDM1-KO, Fig. 3e. Although the cells belonging to the CD4+ Th17-like cluster showed a correlation to IL-10 signaling above, Fig. 2f, the simulated perturbation did not lead to an inversion of its transcriptional program, suggesting that CD4+ Th17 differentiation is not directly driven by IL-10-signaling, Fig. 3f. These findings point back to the subset of CD8+ T cells expressing classical exhaustion signatures. They are highly correlated to IL-10 response leading to further downstream activation of the STAT3-BLIMP-1 axis, as previously reported, detailing cancer-associated T cell dysfunction[12]. Our conclusions raise several questions: What cell types are involved in IL-10 signaling and what is the spatial localization of this interaction. The relevance of this question is highlighted by reports detailing intense crosstalk between tumor, myeloid and lymphoid cells, and improved understanding could significantly contribute to the improvement of therapeutic approaches[5,6,21].

**T cell activation and exhaustion is associated with glioblastoma subtypes.** Glioblastoma has been shown to present a high degree of transcriptional heterogeneity due to regional metabolic differences and varying composition of the tumor microenvironment[29]. To determine the spatial distribution of the above illustrated T cell clusters and its colocalization to defined tumor states, we performed spatially resolved transcriptomic RNA sequencing (stRNA-seq). Tissue samples were obtained from three primary IDH1/2 wildtype glioblastoma patients, and the dataset contained a total of 2352 sequenced spots, Fig. 4a. We observed a median of eight cells per spot (range: 4 to 22 cells per spot), which allows for spatial mapping of gene expression, but not at single-cell resolution[30]. Although this could be considered a limitation, when expression profiles based on the latest glioblastoma subtype classification[29] were spatially mapped, the results confirmed the hypothesized juxta-positioning of the mesenchymal subset to the tumor-associated myeloid compartment[5,21]. These findings were further supported by a recent report that linked myeloid–tumor cell interactions to defined epigenetic immunoediting, resulting in an immunosuppressive phenotype[6]. This would mean that exhausted T cells would be preferentially spatially located within regions of mesenchymal transcriptional programs Fig. 4b. When we spatially correlated our T cell clusters to GBM subtypes using a seeded non-negative matrix factorization (NMF) regression[31] and Moran statistics, tumor regions enriched for mesenchymal-like (MES-like), and astrocytic-like (AC-like) transcriptional signatures were co-localized with activated CD8+ effector, CD8+ T exhausted clusters, and CD4 Th17-like, Fig. 4c, d. To further confirm this spatial dependency, we quantified the putative distance between T cells and GBM subtypes. Spots that were enriched for both MES-like

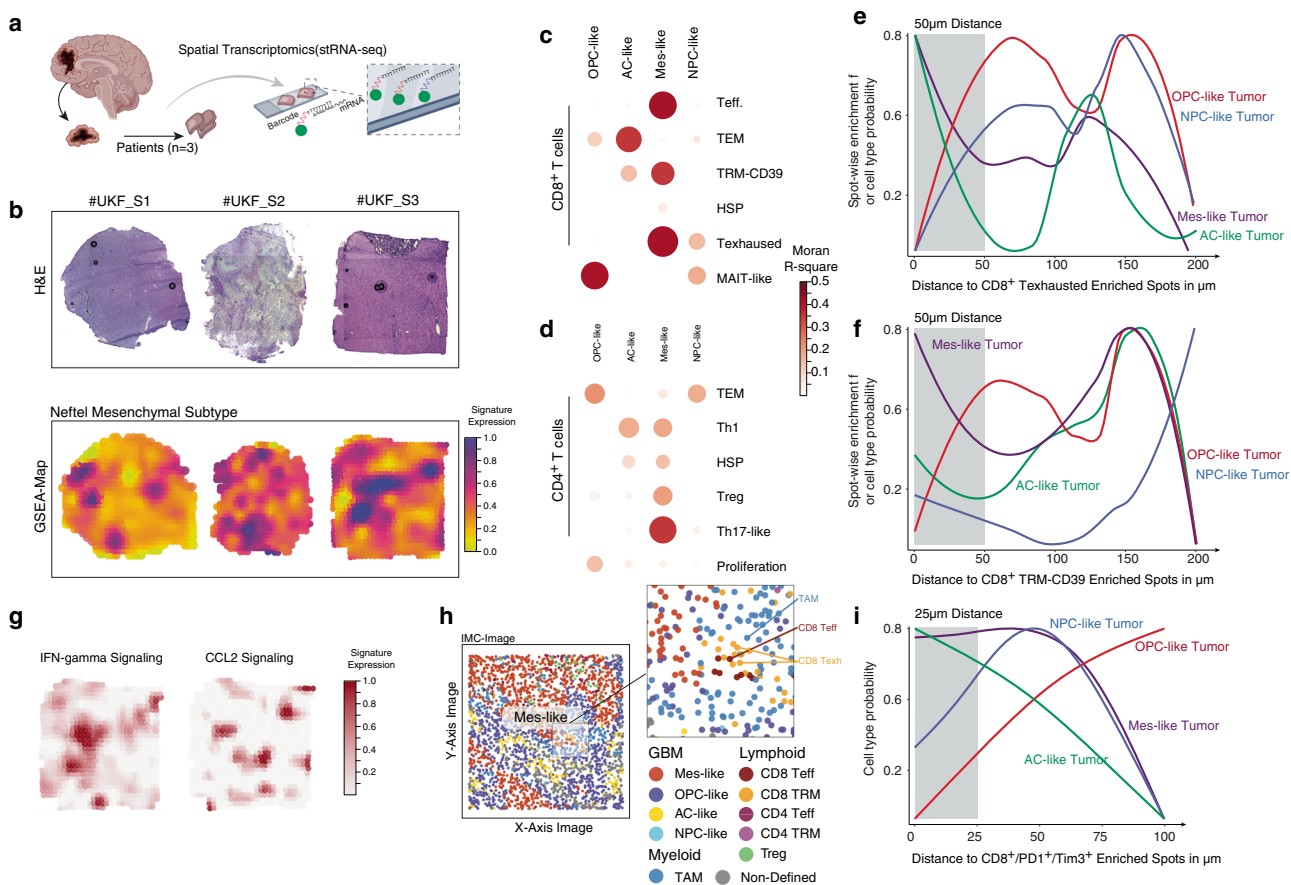

**Fig. 4 Spatial analysis of T cell distribution in glioblastomas. a** Illustration of the workflow using spatially resolved transcriptomics (stRNA-seq). **b** Histological images (H&E), upper plot, and surface plots of the gene set enrichment of the mesenchymal-like signature ($n = 3$ patients) **c**, **d** Spatial correlation analysis with Moran statistics ($R^2$-values) between the GBM subtypes and CD8$^+$ T cell clusters (**c**) and CD4$^+$ T cells (**d**). Point size and color indicate spatial correlation. **e**, **f** Line plots illustrate the distance between the CD8$^+$ clusters (TRM-CD39 (**e**) and T exhausted (**f**)) and tumor subtypes. The x-axis represents the cell type probability computed by gene set enrichment analysis of defined gene sets based on subtype. The y-axis represents the spatial distance to defined T cell clusters based on ranked SPOTlight probabilities. **g** Surface plot of the IFN-gamma and CCL2 response in the S3 tumor sample. **h** Imaging mass cytometry (IMC) data from a GBM sample with annotated cell types. **i** Spatial distance between tumor subtypes and exhausted CD8$^+$ cells from IMC data.

and AC-like signatures were in proximity to exhausted CD8$^+$TRM and T exhausted clusters, suggesting that specific tumor subtypes could be associated with immunosuppression, Fig. 4e, f. The immunosuppressive microenvironment is known to be orchestrated by several cytokines, contributing to the epigenetic immunoediting as recently described[6]. The spatial proximity of *CCL2*, a T cell chemoattractant and IFN-gamma response coupled with the expression of exhaustion markers in MES-like tumor regions was confirmed in the stRNA-seq dataset, Fig. 4g and Supplementary Fig. 4b. Given the lack of single-cell resolution in the stRNA-seq, the dataset was supplemented with imaging mass cytometry (IMC, $n = 1$) to determine precise cellular distances between the CD8$^+$ T cell population (PD1$^+$/TIM3$^+$/CD8A$^+$) and the tumor subgroups (MES-like: EGFR$^+$, CHI3L1$^+$; AC-like: EGFR$^+$, HOPX$^+$; NPC-like: EGFR$^+$, CD24$^+$; OPC-like: EGFR$^+$, OLIG1$^+$), Fig. 4h. The IMC dataset confirmed the spatial juxtaposition of exhausted CD8$^+$ and MES/AC-like tumor cells Fig. 4i.

**CD163$^+$HMOX1$^+$ MΦ's release IL-10 in the tumor microenvironment.** We recently reported that the crosstalk between myeloid cells and reactive astrocytes in the tumor microenvironment was linked to IL10 release, mediated by IFN-gamma-JAK/STAT signaling[19]. To investigate the extent to which the myeloid compartment contributes to the immunosuppressive

dysregulation of T cells, we introduce the "Nearest Functionally Connected Neighbor" algorithm (NFCN). This in silico model was used to identify cell pairs that are most likely to be related, through divergent (up/down) stream signaling activity, Fig. 5a. With our model, cellular interactions with distinct mutual activation requires two fundamental prerequisites. On the one hand, a ligand needs to be expressed and released, or otherwise presented on the cell surface. To minimize the effects of randomly elevated expression or technical artifacts, the occurrence of ligand induction (upstream pathway signaling) was verified. On the other hand, the receiving cell needs to express the receptor, and, as with the inducing cell, downstream signaling has to be activated as well, allowing us to predict the functional status of the receiver cell. To account for the actual physical interaction of the cell pairs estimated to be connected, we calculated the probability of their spatial juxtaposition by integrating stRNA-seq data.

We used our in silico model to screen for potential cells that evoke the IL-10 response in T cells. The algorithm identified pairs of lymphoid (T cell clusters) and myeloid cells (macrophages and microglia cluster) and estimated the likelihood of mutual activation Fig. 5b. Clustering of myeloid cells is presented in the supplementary figure in detail, Supplementary Fig. 5. By extracting the nearest connected cells (top 1% of ranked cells), we identified a subset of myeloid cells characterized by remarkably

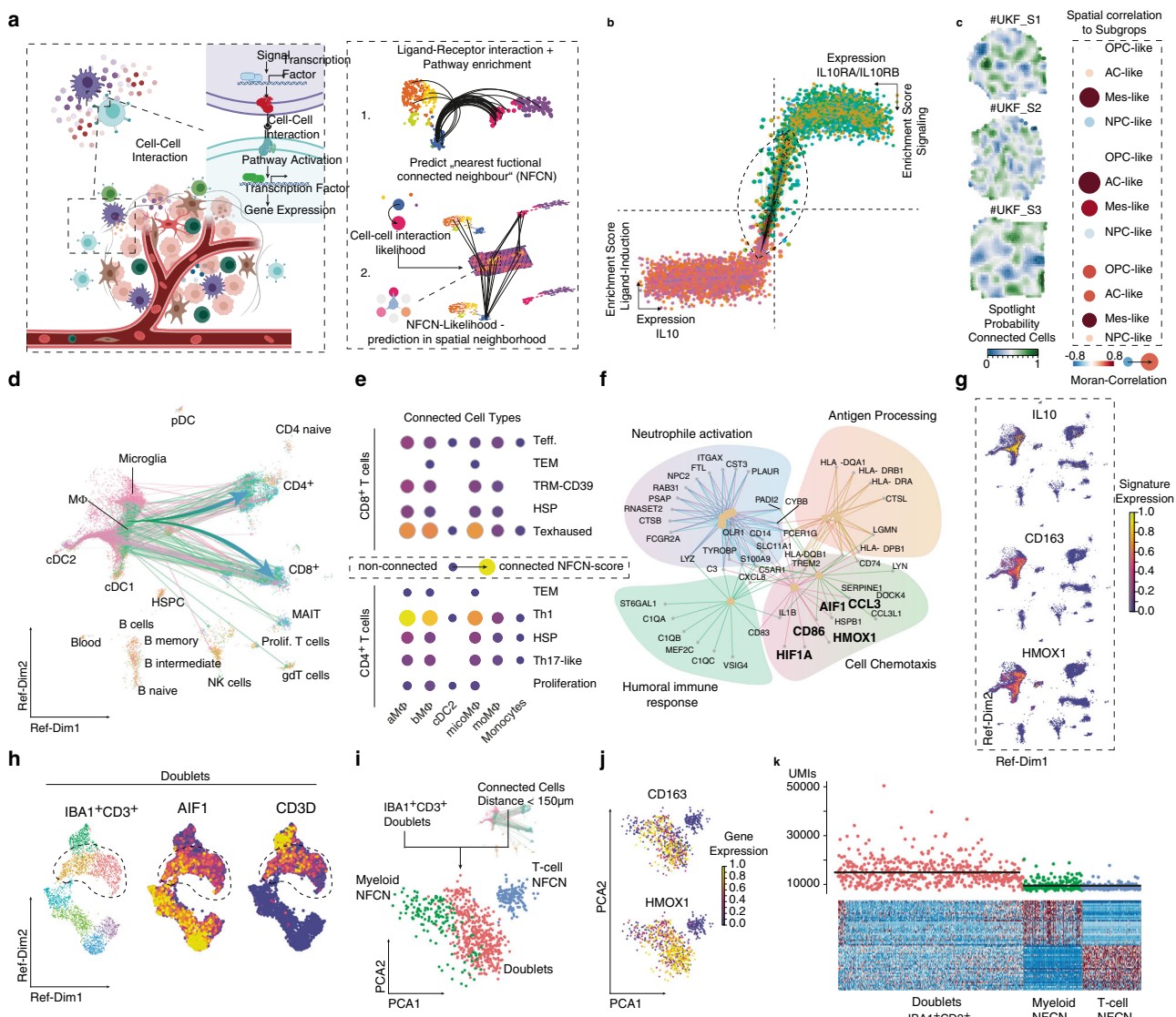

**Fig. 5 Neighborhood analysis of myeloid–lymphoid interactions. a** Workflow exploring cell–cell interactions using two approaches: (1) Predict functional neighbors based on defined ligand-receptor interaction. (2) Validation in spatial transcriptomic datasets. **b** Scatter-cell–cell interaction plot from the "Nearest Functionally Connected Neighbor" algorithm (NFCN) as explained in supplementary results. **c** Spatial alignment of highly connected cell pairs from the NFCN analysis, right side. Spatial correlation analysis is illustrated at the left side. The juxtaposition of connected cells and the GBM subtypes was quantified using Moran statistics. Cells are colored accordingly to the myeloid/T cell cluster of origin. **d** 2D UMAP presentation of the connected cells. Arrows indicate the most likely cell connection between the myeloid and T cell compartment. **e** Dot plot of cluster-wise connectivity of myeloid (on the bottom) and T cell clusters (right side). Color and size indicate the NFCN-score. **f** Gene set enrichment analysis of top five enriched pathways in highly connected myeloid cells. **g** 2D UMAP representation with gene expression. **h** Doublet analysis of mixed CD3+/IBA1+ positive doublets. The dimensional reduction illustrates the identified doublets. The highlighted doublet population demonstrated the CD3+/IBA1+ positive doublets**. i** Fusion of the connected myeloid and lymphoid cells illustrated the gene expression similarity compared to the CD3+/IBA1+ positive doublets. **j** Gene expression of the cells/doublets presented in **i**. **k** Heatmap of the doublets (left) and single cells (right). The UMIs confirmed the status as single cell or doublet. Signature gene expression of either myeloid or lymphoid cell was both expressed in the doublets.

high *IL10* expression. Most of the receiver cells among the connected cells (top 1% of ranked cells) originated from the T cell cluster. This baseline myeloid–lymphoid interaction prediction was further corrected according to their spatial juxtaposition. The most likely position of each ligand- and receptor cell pair was then estimated and putative distance was quantified Fig. 5b. Connected cell pairs with a putative distance >150 μm were excluded from further analysis. The expression signatures from the highly connected cells were then projected onto our stRNA-seq datasets using the SPOTlight[31] algorithm, resulting in a map of IL-10 connectivity. The spatial positioning of the connected cell pairs revealed a significant spatial correlation to MES-like and

AC-like regions of the tumor, Fig. 5c. Estimated cell connections were balanced between CD4+ and CD8+ T cells, Fig. 5d, but mainly included T cells from the CD8+TRM/Exhausted or CD4+Th1 cell clusters, Fig. 5e. We found that myeloid cells from two clusters (aMΦ, bMΦ, Supplementary Fig. 5), characterized by the expression of tumor-associated macrophage genes (*CD163*, *CCL4*, *APOE*, and *HLA-DRA*) were strongly connected to the CD8+ exhausted and CD4+ Th1 T cell clusters. When this myeloid cell population was further characterized using gene set enrichment analysis, significant enrichment of pathways involved in antigen-processing and cytokine signaling was observed, Fig. 5f. This enrichment is coupled with a significant increase in the

expression of Heme Oxygenase 1 (*HMOX1*), activated during inflammation and oxidative injuries, and regulated through the Nrf2/Bach1-axis[32–34], as well as through the IL-10/HMOX1-axis[35]. Furthermore, HMOX1 has been shown to be upregulated in the alternatively activated macrophage subtype[32]. Analysis of the CD163+ HMOX1+ myeloid population revealed a high expression of *IL10*, Fig. 5g. In line with our observations, in the stRNA-seq, *HMOX1* expression was found to be significantly correlated with the expression of mesenchymal-like marker genes, Supplementary Fig. 4c. To further validate potential interactions between the myeloid–lymphoid compartment, we isolated CD3+ IBA1+ (gene: *CD3D* and *AIF1*) doublets from our dataset, Fig. 5h. These doublets contained myeloid–lymphoid cell pairs, assumed to be physically connected[36], Fig. 5i. When comparing the gene expression profile of our top connected cell pairs (Ligand cells: Myeloid cells; Receiver cells: T cells) with the CD3+ IBA1+ doublets, a strong expression of both *CD163* and *HMOX1* was observed, Fig. 5j underpinning the observation that physically connected cells mainly arise from our identified CD163+HMOX1+ myeloid subpopulation, Fig. 5k.

**HMOX1+ IBA1+ myeloid cells are located at the tumor microenvironment interface**. To further evaluate the tumor–host interactions we made use of the recently described human neocortical GBM model, where the cellular architecture of the CNS is well preserved[19,37], Fig. 6a, b. Our data suggest that mesenchymal-like tumor cells are the GBM cells that are involved in the tripartite relationship between myeloid cells, T cells, and GBM cells. Therefore, we inoculated cortical tissue sections with a patient-derived primary GBM cell line (BTSC#233, GFP-tagged, previously characterized by RNA-seq profiling as mesenchymal[38]). The lymphoid compartment was simulated by autografting patient-derived peripheral T cells (CD4+/CD8+ Ratio: 1.2), that were tagged with CellTrace[TM] Far Red (CTFR, Thermo Fischer Scientific) for visualization. The presented model is only a simplification of the complex intercellular dependencies, implying that some confounders were not represented. An obvious aberration from cancer physiology in our model is the absence of bone-derived macrophages. Since the myeloid compartment is predominantly represented by microglia in our model, we aimed to determine whether the mechanism of HMOX activation is shared between macrophages and microglia. This was carried out by reconstructing the spatial juxtaposition, where we found HMOX1+ IBA1+ microglia were present in close proximity to a tumor ($p < 0.023$) and T cells ($p < 0.001$), compared to HMOX1− IBA1+ microglia, Fig. 6c. These findings suggest that the mechanism of HMOX1 activation in macrophages and microglial cells is shared. This ensures the compatibility of the model in the investigation of the effect of HMOX1 expression, even in the absence of bone-derived macrophages. Of note, our results are also in agreement with previous work detailing the characteristic upregulation of HMOX1 in both microglia and macrophages[34]. Since an additional source of bias are other cell types within the microenvironment expressing *HMOX1* (such as astrocytes[39]), we chemically depleted myeloid cells from the model using Clodronate disodium, as previously described[19]. Post depletion, only a minimal number of HMOX1+ cells remained, suggesting that the proportion of non-myeloid HMOX1+ cells represent a negligible minority, Fig. 6d. HMOX1+ IBA1+ myeloid cells were found to be significantly enriched in the direct neighborhood ($<200\,\mu m$) of the tumor-T cell interface, and therefore expected to participate in the hypothesized tripartite relationship, Fig. 6e.

**Myeloid cell depletion results in increased GZMB expression in tumor-infiltrating T cells**. Next, we focused to isolate the effects that HMOX1+IBA1+ myeloid cells have on T cells through IL10-

signaling. We report that the chemical depletion of myeloid cells resulted in a significant reduction of IL10 in the extracellular medium, measured using an enzyme-linked immunosorbent assay (ELISA), Fig. 6f, left. Concurrently, the inflammatory cytokine IL2, which has been reported to be expressed by activated CD4+ and CD8+ T cells[40], was found to be increased after myeloid depletion, Fig. 6f, right. In addition to changes in the cytokine landscape, we report an increased expression of Granzyme B (*GZMB*, marker for effector T cells) in T cells in environments lacking the myeloid compartment, Fig. 6g, h. To evaluate the relationship between GZMB expression and IL-10 levels, the model was treated with an IL-10 neutralizing antibody. When free IL-10 was neutralized, a similar upregulation of GZMB was observed, Fig. 6g. Increased expression of GZMB was coupled with a significant reduction of TIM3 (Gene: *HAVCR2*), Fig. 6g, a characteristic marker for IL-10 affected exhausted T cell cluster, Fig. 1d. Since the regulation of IL-10 release was found to be linked to the JAK-STAT pathway in our recent reports[19], tissue sections were pretreated with Ruxolitinib, an FDA-approved JAK1/2-inhibitor, before inoculation with patient-derived T cells. In line with our above findings, we found strong induction of GZMB expression in T cells, Fig. 6h.

**Case study: JAK-STAT inhibition of a recurrent GBM patient**. With promising results from ex-vivo tissue experiments, a glioblastoma patient, with tumor recurrence after radiation and the CeTeG protocol as well as TTField therapy, was treated in a neoadjuvant setting with Ruxolitinib for 4 weeks as part of the "Compassionate Use" program (RL 2001/83/EG VO 726/2004), before tumor resection, Fig. 7a. Immunostainings of tissue post-resection revealed a significant increase in both CD8+ and CD4+ T cells, with the CD68+ myeloid cell population remaining stable, Fig. 7b, c. When we mapped data from scRNA-seq of CD45+ sorted cells from the treated patient back to our reference dataset, we found that the CD8+/CD4+ balance is markedly shifted in favor of CD8+ cells, Fig. 7d. Reciprocal transformation revealed a shift in T cell populations, with the largest population being T effector and T effector memory cells, suggesting that JAK-inhibition supports T cell activation, Fig. 7e. In the underrepresented CD4+ cell population, the shift towards the effector memory program was even stronger, Fig. 7f. In both CD8+ and CD4+ populations, a highly significant reduction in the number of cells within the exhausted T cell population was observed, suggesting that treatment with JAK-inhibitors could be a potential therapeutic option to boost T cell activation by reducing immunosuppressive programs in both myeloid and glial cells, Fig. 7e, f. Clinically, we observed a strong increase of the contrast-enhanced tumor volume during treatment, which was histologically defined as "pseudoprogression" underpinning the effect of the JAK-inhibitor therapy. Although the treatment history revealed a prognostically poor course with early relapse on therapy (6 months), the disease was stabilized for ~8.5 months after JAK-inhibitor therapy and the patient is still alive after ~24 months, Fig. 7g.

**Discussion**

Although single-cell RNA sequencing is able to accurately map the transcriptional diversity of cellular states[23,29,41,42], spatial information regarding the tissue architecture is lost. Here, we combine single-cell RNA sequencing of the immune compartment along with spatially resolved transcriptomic sequencing (stRNA-seq) to gain spatial insights into the complex crosstalk, cellular states, and cellular plasticity leading to the immunosuppressive environment found in glioblastoma (GBM). Recent studies have reported a myriad of microglia and macrophages

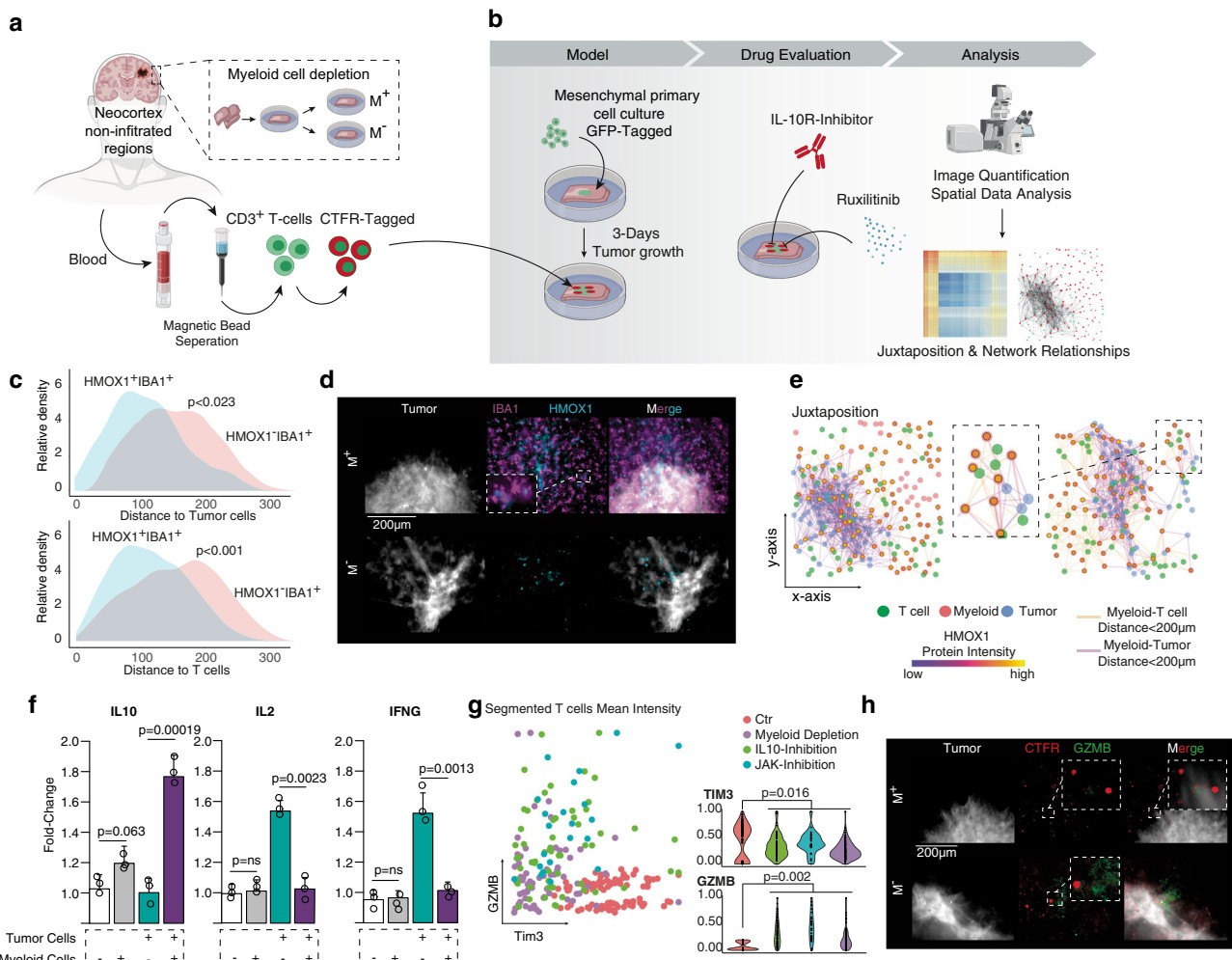

**Fig. 6 Functional modeling of T cell response in human brain slices. a, b** Illustration of the workflow using neocortical slice cultures. **c** Density plots to quantify the spatial distance between HMOX+ and HMOX− IBA1+ myeloid cells and tumor (upper plot) and T cells (bottom). *P* values are determined by non-paired *t*-test statistics after proving the normal distribution. **d** Immunostainings of IBA1 (Macrophages and Microglia) in magenta and HMOX1 in cyan, tumor cells are depicted in gray. In the upper panel, the control set with no myeloid cell depletion (M+) is shown, the bottom panel contains the myeloid cell-depleted sections. The experiments and stainings were repeated at least six times for each condition. **e** Two examples of the reconstruction of cellular relationships. Each point represents a cell as indicated by the legend at the bottom. Lines in orange represent a myeloid-T cell connection (with a distance lower than 100 μm). Tumor-myeloid relationships are illustrated in purple (with a distance lower than 100 μm). The dots of the connected myeloid cells show the expression of HMOX1 by the inner color filling. **f** ELISA measurements of IL-10 and IL-2. *P* values are determined by one-way ANOVA (**c, e, f, j**) adjusted by Benjamini–Hochberg (**c, e, f, j**) for multiple testing. Data were given as mean ± standard deviation. **g** Scatterplot of GZMB (y-axis) versus TIM3 protein level (x-axis). Each dot represents a segmented T cell from different experimental conditions as indicated by the legend at the right. Right part: A quantification of the data. *P* values are determined by one-way ANOVA. **h** Immunostainings of T cells (CSFE-Tagged, in red) and GZMB, a marker of T cell activation (green). The experiments and stainings were repeated at least six times for each condition.

subtypes occupying glioblastoma and other glial tumors[1,4,29,41,42]. However, detailed characterization of the tumor-infiltrating lymphoid compartment is lacking. There has been rising interest in a lymphoid compartment and their varied states due to its importance in furthering our understanding of the immunosuppressive environment of glioblastoma and the development of targeted therapies. T cell states, particularly in disease, are difficult to accurately classify, leading to numerous definitions and associated markers in recent years[2,9,13,43,44]. Some authors use the terms "dysfunctional" and "exhausted" synonymously[45], whereas others differentiate between the dysfunctional and exhausted states of T cells[43,44]. In this study, we use the definition of cellular states proposed by Singer et al., 2016[10]. On the basis of these gene sets, our data showed that only cells which remained chronically activated along the pseudotime trajectory entered a state of dysfunction, and further, exhaustion. The imbalance between pro-

and anti-inflammatory signaling, dominated by IL-10 release, leads to terminal exhaustion of T cells, in agreement with current literature[2,46]. In order to reach a consensus with regard to marker genes, we further validated our findings using a set of marker genes for T cell exhaustion, recently published in an overview study[47]. We and others have previously shown that the GBM microenvironment aids in the evolution of immune suppression. In this process, both astrocytes and myeloid cells, driven by STAT3 signaling, orchestrate the immunosuppressive environment by IL-10 release[4,19,48,49]. Based on the knowledge that IL-10 plays a crucial role in the shift from activation to exhaustion in T cells, we built an in silico model that identified potential cells that drive T cell exhaustion. Using this model, we identified a subset of myeloid cells, marked by high expression of *HMOX1*, the expression of which is induced by oxidative stress and metabolic imbalance[33,34]. *HMOX1* expression is linked to

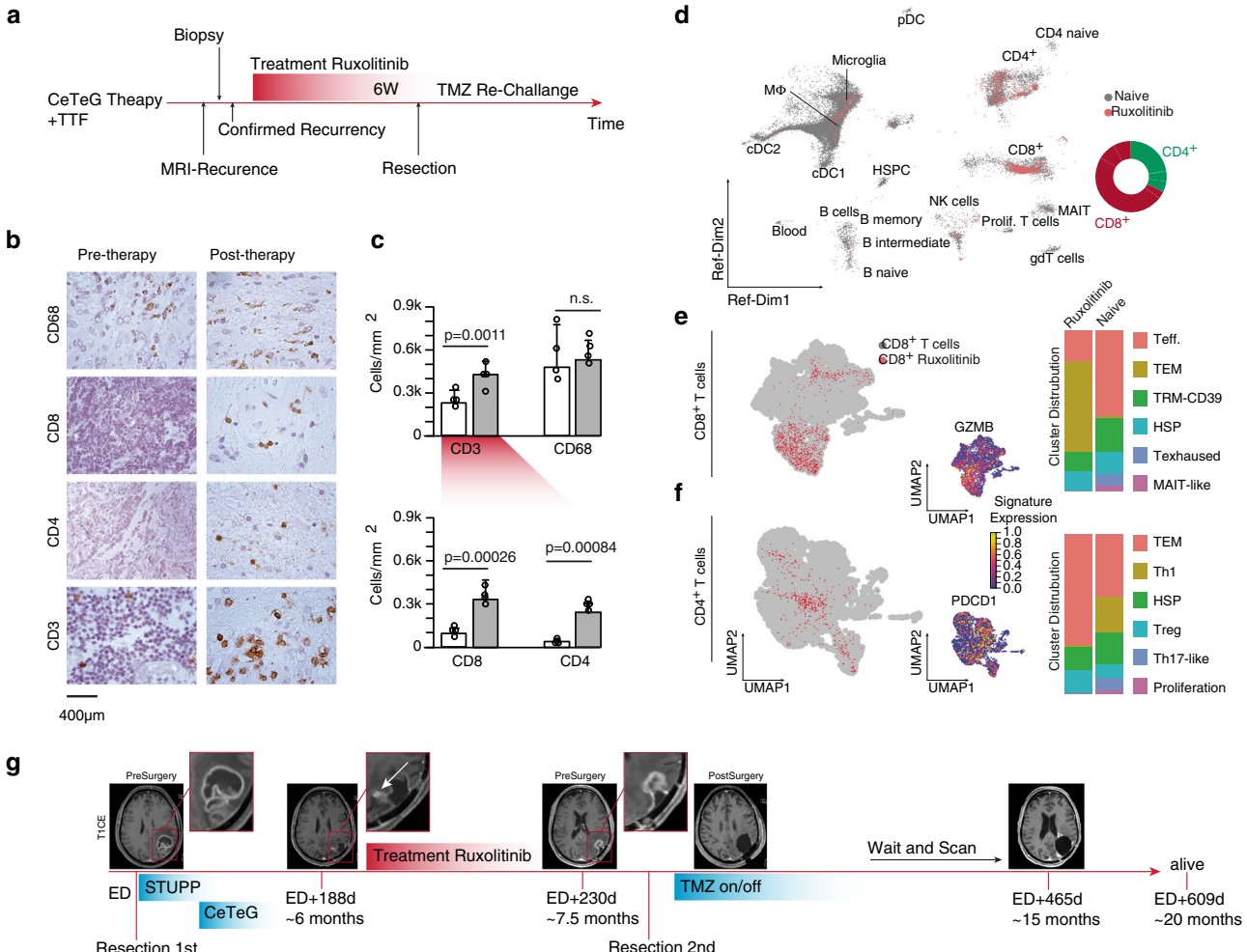

**Fig. 7 JAK-STAT inhibition and T cell response in glioblastoma. a** Timeline of the treatment of our JAK-inhibition GBM case report. **b**, **c** Immunohistochemistry of immune marker and its quantification (**c**), white: pre-therapy, gray: post-therapy, *P* values are determined by one-way ANOVA. **d** Reference UMAP of all T cells (gray) and the JAK-treated patient. A doughnut plot of CD8+/CD4+ positive cells is illustrated on the right side. **e**, **f** Mapping of CD8+ (**e**) and CD4+ (**f**) from the JAK-treated patients to the T cell landscape (left UMAP plot). A quantification is illustrated on the right side. Typical marker gene expression of T effector (GZMB) and exhausted cells (PDCD1) are illustrated at the bottom. **g** Clinical follow-up of the patient with T1-weighted contrast-enhanced MR-images. n.s "not significant".

activation of the STAT3 pathway, which further induces IL-10 production. Furthermore, when we used spatial transcriptomics to identify the spatial overlap of cells that were identified to be highly connected, we saw that the *HMOX1*+ myeloid cells were spatially correlated with T cell exhaustion and the mesenchymal state of glioblastoma. These findings are in agreement with previous reports, revealing that mesenchymal cells are the component of GBM participating in the immune crosstalk[5,21,29]. *HMOX1* expression was found to be increased in recurrence in both GBM and IDH-WT astrocytoma and was negatively associated with overall survival, Supplementary Fig. 6a, b.

In addition, we made use of a human neocortical GBM model coupled with patient-derived T cells to mechanistically validate the role of the myeloid cells with regard to IL-10 release and T cell exhaustion. Fitting with our computational model, we confirmed that the presence of HMOX1+ myeloid cells results in the reduction of the effector T cell population, with a corresponding reduction in IL2 release. This loss of the effector T cell population was coupled with increased expression of the exhaustion marker *TIM3*. Following our recent work where we demonstrated the effectivity of JAK-inhibition in reducing levels of IL-10 in human brain tumors[19], we present that the

inhibition of the JAK-STAT axis was able to partially rescue the immunosuppressive environment, in a single patient. An improvement in the survival of the treated patient compared to patients with a similar prognosis is observed, but permanent disruption of the blood–brain barrier with a repetitive increase of contrast-enhancing lesions was reported. A sampling of these contrast-enhancing lesions showed no evidence of tumor recurrence, suggesting that manipulation of the glia/myeloid environment exacerbated inflammation and resulted in pseudoprogression. Our single-cell RNA-seq confirmed a pronounced enrichment of activated T cells, while the myeloid population remained relatively stable.

In conclusion, this work provides the first glimpse at the lymphocyte population in the glioblastoma microenvironment, where we show that the functional interaction between the myeloid and lymphoid compartment leads to a T cell dysfunction/exhaustion. Using a human neocortical GBM model and single patient subject we showed that this IL-10 driven T cell exhaustion can be rescued by JAK/STAT inhibition. Thus, the results from this work can be used as a stepping stone towards the development of successful immunotherapeutic approaches in the context of GBM.

## Methods

**Ethical approval**. The local ethics committee of the University of Freiburg approved the data evaluation, imaging procedures, and experimental design (protocol 100020/09 and 472/15_160880). The methods used in this work were carried out in accordance with the approved guidelines, with written informed consent obtained from all subjects. The studies were approved by an institutional review board. Further information and requests for resources, raw data, and reagents should be directed and will be fulfilled by the Contact: D. H. Heiland, dieter.henrik.heiland@uniklinik-freiburg.de.

**T-cell isolation and stimulation**. Blood was drawn from a healthy human individual into an EDTA (ethylenediaminetetraacetic acid) blood collection tube. T-cells were extracted by negative selection using a MACSxpress® Whole Blood Pan T-Cell Isolation Kit (Miltenyi Biotec, Bergisch-Gladbach, Germany). T-cells were then transferred to Advanced RPMI 1640 Medium (Thermo Fisher Scientific, Carlsbad, USA) and split for cytokine treatment: Three technical replicates were used for each T-cell treatment condition. Interleukin 2 (IL-2, Abcam, Cambridge, UK) was used at a final concentration of 1 ng/ml, Interleukin 10 (IL-10, Abcam, Cambridge, UK) at 5 ng/ml, Interferon gamma (IFN-γ, Abcam, Cambridge, UK) at 1 ng/ml, and Osteopontin (SPP-1, Abcam, Cambridge, UK) at 3 μg/ml. Cytokine treatment was performed in Advanced RPMI 1640 Medium at 37 °C and 5% $CO_2$ for 24 h.

**RNA sequencing of stimulated T-cells**. Purification of mRNA from total RNA samples was carried out using the Dynabeads mRNA Purification Kit (Thermo Fisher Scientific, Carlsbad, USA). The subsequent reverse transcription reaction was performed using SuperScript IV reverse transcriptase (Thermo Fisher Scientific, Carlsbad, USA). RNA sequencing preparation was carried out using, the Low Input by PCR Barcoding Kit and the cDNA-PCR Sequencing Kit (Oxford Nanopore Technologies, Oxford, United Kingdom), using the MinION Sequencing Device, the SpotON Flow Cell (R9.4.1), and MinKNOW software (Oxford Nanopore Technologies, Oxford, United Kingdom) according to the manufacturer's instructions. Samples were sequenced for 48 h on two flow cells. Base-calling was performed by Albacore implemented in the nanopore software. Only $D^2$-Reads with a quality score above 8 were used for further alignment.

**Sequence trimming and alignment**. We made use of our previously reported[19] automated pipeline for nanopore cDNA-seq data, available at GitHub (https://github.com/heilandd/NanoPoreSeq). Multiplexed samples were separated by their barcodes and trimmed using Porechop (https://github.com/rrwick/Porechop). Alignment was carried out using minimap2[50] and processed with sam-tools[51].

**Post hoc analysis of bulk RNA sequencing data**. A matrix of genes counts was further processed using the RawToVis.R script (github.com/heilandd/Vis_Lab1.5), normalizing of mapped reads by DESeq2[52], batch effect removal using ComBat[53] and fitting for differential gene expression. Gene set enrichment analysis was performed by transforming the log2 fold change of DE into a ranked $z$-scored matrix, using clusterProfiler[54,55]. The expression matrix was analyzed using AutoPipe (https://github.com/heilandd/AutoPipe), a supervised machine-learning algorithm and visualized using a heatmap, implemented in the stats package in R. Final visualization was carried out using Visualization package (Vis_Lab, github.com/heilandd/Vis_Lab_1.5), implemented in R as a shiny dashboard app. The top 50 up/downregulated genes of each stimulation condition with respect to control was used to construct the stimulation library.

**Single-cell suspensions for scRNA-sequencing**. Tumor tissue was obtained from glioma surgery immediately after resection and was transported in phosphate-buffered saline (PBS) within ~5 min into our cell culture laboratory. Resected tissue was reduced to small pieces under sterile conditions and was further processed using the Neural Tissue Dissociation Kit (T) using C-Tubes (Miltenyi Biotech, Bergisch-Gladbach, Germany) according to the manufacturer's instructions. Residual myelin and extracellular debris was eliminated using the Debris Removal Kit (Miltenyi Biotech, Bergisch-Gladbach, Germany). Erythrocytes were removed by resuspending and incubating the obtained pellet in 3,5 ml ACK-lysis buffer (Thermo Fisher Scientific, Carlsbad, USA) for 5 min, followed by centrifugation (350×$g$, 10 min, RT). Cell counts were quantified using a hematocytometer after resuspending the pellet in PBS. The final cell suspensions were centrifuged again (350×$g$, 10 min, RT) and resuspended in a freezing medium containing 10% DMSO (Sigma-Aldrich, Schnelldorf, Germany) in FCS (PAN-Biotech, Aidenbach, Germany). Cell suspensions were immediately placed in a storage container (Mr. Frosty™, Thermo Fisher Scientific, Carlsbad, USA) and stored at −80 °C freezer for not more than 4 weeks.

**Sample preparation for scRNA-sequencing**. Four single-cell suspensions, originating from one patient with an IDH-mutated glioma and three patients with an IDH-wildtype glioblastoma (GBM), were thawed and dead cells magnetically labeled and eliminated using a Dead-Cell Removal Kit (Miltenyi Biotech). The tumor immune compartment, T-cells in particular were positively selected by using

CD3+ conjugated magnetic beads (Miltenyi Biotech, Bergisch-Gladbach, Germany). Cells were stained with trypan blue, counted using a hematocytometer, and prepared at a concentration of 700 cells/μL.

**Droplet-based scRNA-sequencing**. Around 16,000 cells per sample were loaded on the Chromium Controller (10x Genomics, Pleasanton, CA, USA) per Chromium Next GEM Single Cell 3´v3.1 reaction (10x Genomics, Pleasanton, CA, USA), for droplet-based scRNA-sequencing. Library construction and sample indexing was carried out according to the manufacturer´s instructions. scRNA-libraries were sequenced on a NextSeq 500/550 High Output Flow Cell v2.5 (150 Cycles) on an Illumina NextSeq 550 (Illumina, San Diego, CA, USA). The bcl2fastq function provided by Illumina and the cell ranger (v3.0) function provided by 10x Genomics were used for quality control.

**Postprocessing of scRNA-sequencing data**. Cell ranger (10x Genomics) was used to detect low-quality read pairs from single-cell RNA sequencing (scRNA-seq) data. Reads that did not satisfy the following criteria were filtered out: (1) Bases with quality <10, (2) no homopolymers, and (3) "N" bases accounting for ≥10% of the read length. The filtered reads were then mapped using STAR aligner[56] and the resulting filtered count matrix was further processed using Seurat v3.0[57] in the R computing environment. Gene expression values were normalized by dividing each estimated cell by the total number of transcripts and multiplied by 10,000, followed by a natural-log transformation. Batch effects were then removed, and the data were scaled using a regression model including the sample batch information and percentage of ribosomal and mitochondrial gene expression. For downstream analysis, we made use of 2000 most variably expressed genes and the decomposed eigenvalue frequencies of the first 100 principal components, determined by the number of nontrivial components in comparison to randomized expression values. These nontrivial components were then used for SNN clustering[58] followed by dimensional reduction using the UMAP algorithm. Differentially expressed genes (DE) of each cluster were obtained using a hurdle model tailored to scRNA-seq data, part of the MAST software package (https://github.com/RGLab/MAST). Cell types were identified using three different methods; Classical expression of signature markers of immune cells; SingleR an automated annotation tool for single-cell RNA sequencing data obtaining signatures from the Human Primary Cell Atlas, SCINA, a semi-supervised cell type identification tool using cell-type signatures as well as a gene set variation analysis (GSVA). Obtained results were combined and clusters were assigned to cell type based on highest enrichment within all models. In order to individually analyze T-cells, we used the assigned cluster and filter for the following criteria: CD3+ CD8+/CD4+ CD14− LYZ− GFAP− CD163− IBA−.

**Spatial transcriptomics**. Spatial resolved transcriptomic data was acquired using the Spatial transcriptomics kit (10x Genomics, Pleasanton, CA, USA). Tissue Optimization and Library preparation were carried out according to the manufacturer's protocol, as described below.

**Tissue collection and RNA quality control**. Tissue samples from three patients, diagnosed with WHO IV glioblastoma multiforme (GBM), were included in this study. Fresh tissue collected immediately post-resection was rapidly embedded in optimal cutting temperature compound (OCT, Sakura, Japan) and snap-frozen in liquid $N_2$. The prepared tissue samples were stored at −80 °C until further processing. A total of ten sections (10 μm each) per sample were lysed using TriZOl (15596026, Invitrogen, Thermo Fisher Scientific, Carlsbad, USA) and used to determine RNA integrity. Total RNA was extracted using the PicoPure RNA Isolation Kit (KIT0204, Thermo Fisher Scientific, Carlsbad, USA) according to the manufacturer's protocol. RIN values were determined using a 2100 Bioanalyzer (RNA 6000 Pico Kit, Agilent Technologies, Santa Clara, CA, USA) according to the manufacturer's protocol.

**Tissue staining and imaging**. 10 μm tissue sections were mounted onto spatially barcoded glass slides with poly-T reverse transcription primers, one section per array. The prepared slides were warmed to 37 °C, following which the sections were fixed using 4% formaldehyde solution (P087.1, Carl Roth, Karlsruhe, Germany) for 10 min, which was then rinsed using PBS. The fixed sections were covered with 2-Propanol (20842312, VWR International, Radnor, PA, USA). Post evaporation for 40 s, sections were incubated in Mayer's Hematoxylin (1092490500, VWR International, Radnor, PA, USA) for 7 min, Dako bluing buffer (CS70230-2, Agilent Technologies, Santa Clara, CA, USA) for 90 s, and finally in Eosin Y (E4382, MilliporeSigma, St. Louis, MO, USA) for 1 min. The glass slides were then washed with RNase/DNase-free water and incubated at 37 °C for 5 min or until dry. Before imaging, the glass slides were mounted with 87% glycerol (A3739, AppliChem, Darmstadt, Germany) and covered with coverslips (01-2450/1, R. Langenbrinck, Emmendingen, Germany). Brightfield imaging was performed at 10x magnification with an inverted microscope (Axio Imager 2, Zeiss, Jena, Germany), post-processed using ImageJ software. Post Imaging, coverslips, and glycerol were removed by washing the glass slides in RNase/DNase-free water, after which the slides were washed using 80% ethanol to remove final traces of glycerol.

**Permeabilization, cDNA synthesis, and tissue removal.** For each capture array, 70 μL of pre-permeabilization buffer, containing 50 U/μl Collagenase along with 0.1% Pepsin in HCl was added, incubated for 20 min at 37 °C. Each array well was then carefully washed using 100 μl 0.1x SSC buffer, following which 70 μl of Pepsin was added and further incubated for 11 min at 37 °C. Each well was then washed as previously described and 75 μl of cDNA synthesis master mix containing: 96 μl of 5X First-strand buffer, 24 μl 0.1 M DTT, 255.2 μl of DNase/RNase free water, 4.8 μl Actinomycin, 4.5 μl of 20 mg/ml BSA, 24 μl of 10 mM dNTP, 48 μl of Superscript®, and 24 μl of RNAseOUT™ was added to each well and incubated for 20 h at 42 °C. Cyanine-3-dCTP was used to aid in the determination of the footprint of the tissue section used.

Since glioblastoma tissue is fatty tissue, degradation and tissue removal was carried out using Proteinase K, where 420 μl Proteinase K and PKD buffer (1:7), were added to each well and incubated at 56 °C for 1 h with intermittent agitation (15 s/3 min). After incubation, the glass slides were washed with 100 ml of 50 °C SSC/SDS buffer with agitation three times, for 10 min, 1 min, and finally for 1 min at 300 rpm. The glass slides were then air-dried at RT. Tissue cleavage was carried out by the addition of 70 μl of cleavage buffer (320 μl RNase/DNase-free water, 104 μl Second strand buffer, 4.2 μl of 10 mM dNTP, 4.8 μl of 20 mg/ml BSA, and 48 μl of USER™ Enzyme) to each well and the slides were incubated at 37 °C for 2 h with intermittent agitation.

**Spot hybridization.** In order to determine the exact location and quality of each of the 1007 spots, fluorescent Cyanine-3 A is hybridized to the 5′ ends of the surface probes. About 75 μL of the hybridization solution (20 μl of 10 μM Cynaine-3 A probe and 20 μl of 10 μM Cyanine-3 Frame probe in 960 μl of 1X PBS) was added to each well, incubated for 10 min at RT. The slides were then washed thrice with 100 ml of SSC/SDS buffer preheated to 50 °C for 10 min, 1 min, and 1 min at RT with agitation. The slides were then air-dried and imaged after application of Slowfade® Gold Antifade mountant (S36936, Thermo Fisher Scientific, Carlsbad, USA) and a coverslip.

**Library preparation**

*Second strand synthesis.* About 5 μl second strand synthesis mix containing 20 μl of 5X First-Strand Buffer, 14 μl of DNA polymerase I (10 U/μl), and 3.5 μl Ribonuclease H (2 U/μl) were added to the cleaved sample and incubated at 16 °C for 2 h. Eppendorf tubes were placed on ice and 5 μl of T4 DNA polymerase (3 U/μl) were added to each strand and incubated for 20 min at 16 °C. About 25 μl of 80 mM EDTA (30 μl 500 mM EDTA with 158 μl DNase/RNase free water) was added to each sample and the samples were kept on ice.

*cDNA purification.* cDNA from the previous step was purified using Agencourt RNAclean XP beads (Beckman Coulter, Brea, CA, USA) in a DynaMag™- 2 magnetic rack (12321D, Thermo Fisher Scientific, Carlsbad, USA), incubated at RT for 5 min. Further cleansing was carried out by the addition of 80% ethanol, while the samples were still placed in the magnetic rack. Sample elution was then carried out using 13 μl of NTP/water mix.

*In vitro transcription and purification.* cDNA transcription to aRNA was carried out by the addition of 4 μl of reaction mix containing: 10x Reaction Buffer, T7 Enzyme mix, and SUPERaseIn™ RNase Inhibitor (20U/μL) to 12 μl of eluted cDNA, incubated at 37 °C, for 14 h. Samples were purified using RNAclean XP beads according to the manufacturer's protocol and further eluted into 10 μL DNase/RNase free water. The amount and average fragment length of amplified RNA was determined using the RNA 6000 Pico Kit (Agilent, 5067-1513) with a 2100 Bioanalyzer according to the manufacturer's protocol.

*Adapter ligation.* Next, a 2.5 μL Ligation adapter (IDT) was added to the sample and was heated for 2 min at 70 °C and then placed on ice. A total of 4.5 μL ligation mix containing 11.3 μL of 10X T4 RNA Ligase, T4 RNA truncated Ligase 2, and 11.3 μL of murine RNase inhibitor was then added to the sample. Samples were then incubated at 25 °C for 1 h. The samples were then purified using Agencourt RNAclean XP beads (Beckman Coulter, Brea, CA, USA) according to the manufacturer's protocol.

*Second cDNA synthesis.* Purified samples were mixed with 1 μl cDNA primer (IDT), 1 μl dNTP mix up to a total volume of 12 μl, and incubated at 65 °C for 5 min and then directly placed on ice. A 1.5 ml Eppendorf tube containing 8 μl of the sample was mixed with 30 μl of 5X First-Strand Buffer, 7.5 μl of 0.1 M DTT, 7.5 μl of DNase/RNase free water, 7.5 μl of SuperScript® III Reverse transcriptase, and 7.5 μl of RNaseOUT™ Recombinant ribonuclease Inhibitor, incubated at 50 °C for 1 h followed by cDNA purification using Agencourt RNAClean XP beads (Beckman Coulter, Brea, CA, USA) according to the manufacturer's protocol. Samples were then stored at −20 °C.

*PCR amplification.* Prior to PCR amplification, we determined that 20 cycles were required for appropriate amplification. A total reaction volume of 25 μl containing 2x KAPA mix, 0.04 μM PCRInPE2 (IDT), 0.4 μM PCR InPE1.0 (IDT), 0.5 μM PCR Index (IDT), and 5 μL of purified cDNA were amplified using the following

protocol: 98 °C for 3 min followed by 20 cycles at 98 °C for 20 s, 60 °C for 30 s, 72 °C for 30 s followed by 72 °C for 5 min. Libraries were purified according to the manufacturer's protocol and eluted in 20 μL elution buffer. The samples were then stored at −20 °C until used.

*Quality control of libraries.* The average length of the prepared libraries was quantified using an Agilent DNA 1000 high sensitivity kit, using a 2100 Bioanalyzer (Agilent Technologies, Santa Clara, CA, USA). The concentration of the libraries was determined using a Qubit (dsDNA High Sensitivity kit, Thermo Fisher Scientific, Carlsbad, USA). Libraries were diluted to 4 nM, pooled, and denatured before sequencing on the Illumina NextSeq platform using paired-end sequencing. We used 30 cycles for read 1 and 270 cycles for read 2 during sequencing (Primer sequences in the source data file).

**Postprocessing spatial transcriptomics data.** The H&E staining image was aligned using the manufacturer-provided st-pipeline (github.com/SpatialTranscriptomicsResearch/st_pipeline). The pipeline contains the following steps: Quality trimming and removing of low-quality bases (bases with quality <10), removal of homopolymers, normalization of AT and GC content, mapping read 2 using STAR, demultiplexing based on read 1, sort for reads (read1) with valid barcodes, annotation of reads using htseq-count, group annotated reads by barcode position, gene and genomic location (with an offset) to get a read count (github.com/SpatialTranscriptomicsResearch/st_pipeline). The output from the pipeline consists of a gene count matrix, a spatial information file containing the x and y coordinates of each spot, and the H&E image. As described for the scRNA-seq, Seurat v3.0 was used to normalize gene expression values, regression of cell cycle effects, determination of nontrivial components, SNN clustering, and finally dimensional reduction and visualization using the UMAP algorithm. We further built and provide a user-friendly viewer for spatial transcriptomic data and provide tutorials on analysis of data: https://themilolab.github.io/SPATA/.

**Spatial gene expression.** For spatial expression plots, we used either normalized and scaled gene expression values (for single genes) or enrichment scores of defined gene sets, using the 0.5 quantiles of a probability distribution fitting. The x-axis and y-axis coordinates are obtained from an input file based on the localization of the H&E staining. We computed a matrix based on the minimum and maximum extension of the spots used ($32 \times 33$) containing either gene expression or computed scores. Spots not covered by tissue were set to zero. The matrix was then transformed using the squared distance between two points divided by a given threshold, implemented in the fields package (https://github.com/dnychka/fieldsRPackage) and the input values were adapted by increasing the contrast between empty spots. The data were represented either as surface plots using the Plotly package[59] or as images using the graphics package, both implemented in the R computing environment.

**Representation of cellular states.** We aligned spots to variable states based on predefined gene sets (GS) that were selected $GS_{(1,2,..n)}$. First, we spots were separated into $GS_{(1+2)}$ versus $GS_{(2+4)}$, using the following equation:

$$A_1 = \| \, GS_{(1)}, GS_{(2)} \|_\infty - \| \, GS_{(3)}, GS_{(4)} \|_\infty \qquad (1)$$

A1 defines the y-axis of the two-dimensional representation. In a next step, we calculated the x-axis separately for spots A1 < 0 and A1 > 0:

$$A1 > 0: A_2 = \log 2 \, (\overline{GS_{(1)}} - [\overline{GS_{(2)}} + 1]) \qquad (2)$$

$$A1 < 0: A_2 = \log 2 \, (\overline{GS_{(3)}} - [\overline{GS_{(4)}}]) \qquad (3)$$

For further visualization of the enrichment based on gene set enrichment across the two-dimensional representation, we transformed the distribution to representative colors using a probability distribution fitting. This representation is an adapted method published by Neftel and colleagues recently[29,42].

**Spatial correlation analysis.** Spatial correlation analysis was carried out by performing background noise reduction using a deep autoencoder followed by a Bayesian correlation model. In a first step, noise reduction was carried out using an autoencoder similar to that recently described for single-cell RNA sequencing studies[60]. The autoencoder comprises of both an encoder and decoder component, which can be defined as transitions:

$$\text{encoder}: \phi : \mathscr{X} \to \mathscr{F} \quad \text{decoder}: \psi : \mathscr{F} \to \mathscr{X} \qquad (4)$$

$$\phi, \psi = \arg.\min \| X - (\phi, \psi) X \|^2 \qquad (5)$$

The encoder stage takes the input $x \in \mathbb{R}^d = \mathscr{X}$ and maps it to $z \in \mathbb{R}^p = \mathscr{F}$ at the layer position $\varphi$:

$$x = A^{\varphi=0}; z^\varphi = \mathbf{ReLU}(W^\varphi \times A^{\varphi-1} + b^\varphi) \qquad (6)$$

$z^\varphi$ is also referred to as latent representation, here presented as $z^1, z^2, \ldots, z^{\varphi=n}$, where $\varphi$ describes the number of hidden layers. **W** is the weight matrix and **b** represent the dropout/bias vector. The network architecture contained 32 hidden layers, as recommended[60]. In the decoder, weights and

biases are reconstructed through backpropagation ($\psi : \mathcal{F} \to \mathscr{X}$) and z is mapped to $x' = A^{0'}$, in the shape as $x'$.

$$A^{\varphi-1'} = \sigma'(W^{\varphi'} \times z^{\varphi} + b^{\varphi'}) \tag{7}$$

At this stage, $W', \sigma', b'$ from the decoder are unrelated to $W, \sigma, b$ from the encoder. Therefore, we used a loss function to train the network to minimize reconstruction errors.

$$\mathcal{L}(x, x') = x - \sigma'(W'(\sigma(Wx + b)) + b')^2 \tag{8}$$

In the second step, we used the predicted gene expression matrix ($x'$) and fitted and Bayesian correlation model[61] (https://github.com/rasmusab/bayesian_first_aid). An illustration of spatial correlation is provided in Supplementary Fig. 7.

**Integration of scRNA sequencing data into a spatial context**. In order to integrate the identified cellular clusters into the cartesian space of spatially resolved transcriptomic data, we used the recently published Spotlight algorithm, integrating the output into our SPATA objects for visualization[31].

**RNA-velocity and pseudotime trajectory analysis**. In order to determine dynamic changes in gene expression, we extracted splicing information from the *.bam files generated by cell ranger, using the velocyto.py tool[62]. The resulting *.loom files were merged and transformed into h5ad format for further processing by scVelo[63] and CellRank[64]. This analysis pipeline is integrated into the SPATA toolbox[65]. Single-cell data was used to generate a SPATA S4 object, using the UMAP coordinates as spatial coordinates. The output of the scVelo script (implemented in the development branch of the SPATA toolbox) was imported into the SPATA S4 object (*slot:@fdata*) and for visualization. RNA-velocity streams were converted into trajectories and also imported into the SPATA S4 object (*slot:@trajectories*). Dynamic gene expression changes along trajectories was performed using the "assessTrajectoryTrends()" function within SPATA.

**Gene set enrichment analysis**. Gene sets were obtained from the MSigDB v7 database[66] and was supplemented with internally created gene sets, available at github.com/heilandd. Normalized and centered expression data which was further transformed to z-scores ranging from 1 to 0 was used for enrichment analysis of single clusters.

Genes were ranked by log fold change which was then used as the input for Gene Set Enrichment Analysis. For IL10 signaling we used the signatures: "BIOCARTA_IL10_PATHWAY", "GOBP_INTERLEUKIN_10_PRODUCTION", "REACTOME_INTERLEUKIN_10_SIGNALING". For TGB beta signaling: "BIOCARTA_TGFB_PATHWAY".

**Identification of cycling cells**. Proliferation scores were calculated based on genesets generated by Neftel. et. al., using the Gene Set Variation Analysis (GSVA) software package[67] in the R computing environment. The analysis was based on a nonparametric unsupervised approach, transforming a classic gene matrix (gene-by-sample) into a gene set by sample matrix, resulting in an enrichment score for each sample and pathway. From the calculated enrichment scores, a threshold was determined based on distribution fitting to define cycling cells.

**Cell type prediction**. Meaningful components ($n = 54$) were generated from the eigenvalue frequencies of the first 100 principal components. Shared-nearest neighbor (SNN) graph clustering resulted in 21 clusters (C0–C20) containing uniquely gene expression profiles. The major observed cell type when using the semi-supervised subtyping algorithm of scRNA-seq (SCINA-Model)[68] and SingleR[69] are: microglia cells (*TMEM119, CX3CR1,* and *P2RY12*) and macrophages (*AIF1, CD68, CD163* and low expression of *TMEM119, CX3CR1*), followed by CD8$^+$ T-cells (*CD8A* and *CD3D*), natural killer cells (*KLRD1, GZMH, GZMA, NKG7,* and *CD52*), CD4$^+$ T-cells (*BCL6, CD3D, CD4, CD84,* and *IL6R*), T-memory cells (*TRBC2, LCK, L7R,* and *SELL*), granulocytes (*LYZ*), a minor number of oligodendrocytes and oligodendrocyte-progenitor cell (OPC's) (*OLIG1, MBP,* and *PDGFA*), and endothelial cells (*CD34, PCAM1,* and *VEGFA*) Fig. 1b and Supplementary Fig. 1b–f.

**Nearest functionally connected neighbor (NFCN) algorithm**. In the scope of this work, we created a model: Nearest Functionally Connected Neighbor (NFCN), to identify connected cells that interact by means of defined activation or inhibition of downstream signaling in the responder cell. The model assumes that a cell–cell interaction is only when a receptor/ligand pair induces corresponding downstream signaling within the responder cell (cell with expressed receptor). Furthermore, we consider that the importance of an activator cell (cell with expressed ligand) can be ranked according to their enriched signaling, responsible for inducing expression of the ligand. Based on these assumptions, we defined an algorithm to map cells along an interaction trajectory. The algorithm was designed to identify potential activators from defined subsets of cells.

A normalized and scaled gene expression matrix, a string containing the subset of target cells, a list of genes defining ligand induction on the one side, and receptor

signaling on the other side are used as input for the algorithm. Genes were chosen from the MSigDB v7 database or our own stimulation library, generated as described previously. We then downscaled the data to 3000 representative cells including all myeloid cell types, calculating the enrichment of induction and activation of the receptor/ligand pair of interest. Enrichment scores were calculated by singular value decomposition (SVD) over the genes in the gene set, with the coefficients of the first right-singular vector defining the enrichment of induction/activation profiles. Both expression values and enrichment scores were fitted using a probability distribution model, with cells outside the 95% quantile rejected. We then fitted a model using a nonparametric kernel estimation (Gaussian or Cauchy-Kernel), based on receptor/ligand expression ($A_{exp}$) and up/downstream signaling ($A_{eff}$) of each cell (i = {1,..n}):

$$n_{\exp i} = \frac{A_{\exp i} - \min(A_{\exp})}{\max(A_{\exp}) - \min(A_{\exp})} \tag{9}$$

$$\widehat{f}_h(n_{\exp i}) = \frac{1}{n}\sum_{i=1}^{n} K_h(n_{\exp} - n_{\exp i}) \tag{10}$$

Where K is the kernel and $0.7 > h > 0.3$ was used to adjust the estimator. The model resulted in a trajectory defined as Ligand$^{(-)}$-Induction$^{(-)}$ to cells of the target subset with Receptor$^{(-)}$-Activation$^{(-)}$. Further, cells were aligned along the "interaction trajectory". We defined connected cells as those falling in the upper 70% CI in receptor/ligand expression as well as sores of induction/activation. The representation process is illustrated schematically in Fig. 5a. In addition, receptor–ligand interactions were also determined using NicheNet[70].

**CNV estimation**. Copy-number variations (CNVs) were estimated by aligning genes to their chromosomal location and applying a moving average to the relative expression values, with a sliding window of 100 genes within each chromosome, as described recently[23]. First, we arranged genes in accordance with their respective genomic localization using the CONICSmat software package[71]. As a reference set of non-malignant cells, we identified and used 400 CD8$^+$ cells (low likelihood of expression by tumor cells). To avoid the biasing the moving average based on the expression of specific genes, we limited relative expression values [−2.6, 2.6], replacing all values above/below $\exp_{(i)} = |2.6|$ using the infercnv software package (https://github.com/broadinstitute/inferCNV), as previously reported[15].

**Flow cytometry**. Single-Cell suspensions were cleared of dead cells using the Dead-Cell Removal Kit (Miltenyi Biotech, Bergisch-Gladbach, Germany) and enriched for CD3$^+$ cells by CD3-MACS enrichment (Miltenyi Biotech, Bergisch-Gladbach, Germany). Isolated cells were then incubated with VivaFix™ 398/550 (BioRad Laboratories, CA, USA) according to the manufacturer´s instructions, followed by fixation in 4% paraformaldehyde (PFA) for 10 min. After centrifugation ($350\times g$; 4 °C; 5 min), the cell pellet was suspended in 0.5 ml 4 °C FACS buffer. Cell suspensions were then centrifuged at $350\times g$ for 5 min, followed by resuspension in FACS buffer, twice. Finally, cells were resuspended in 0.5–1 ml of FACS buffer, depending on cell count. Sorting was carried out using a Spectral cell Analyzer (SP6800, Sony Biotechnology, CA, USA) in standardization mode, with photomultiplier tube voltage set to maximum in order to reach a saturation rate below 0.1%. Gating was performed using FCS Express 7 plus (De Novo Software, CA, USA) at the Lighthouse Core Facility, University of Freiburg.

**Immunofluorescence**. Culture media was removed from wells containing tissue sections on insets. Tissue sections were fixed using 1 ml of 4% paraformaldehyde (PFA) per well for 1 h, followed by incubation in 20% methanol for 5 min. Permeabilization was carried out using 1% Triton (X-100, MilliporeSigma, St. Louis, MO, USA) overnight at 4 °C and further blocked using 20% BSA for 4 h. The permeabilized and blocked sections were then labeled with primary antibodies in 5% BSA-PBS incubated overnight at 4 °C. Washed sections were labeled with secondary antibodies conjugated with Alexa 405, 488, 555, or 568 for 3 h at RT. Finally, sections were mounted on glass slides using DAPI fluoromount (0100-20, Southern Biotech, AL, USA), as previously described[19].

**Human organotypic tissue culture**. Human neocortical tissue cultures were prepared as recently described[19,37,72]. In brief, damaged tissue was dissected away from the tissue block in a preparation medium containing Hibernate medium supplemented with 13 mM Glucose and 30 mM NMDG. About 300-μm-thick coronal tissue sections were generated using a vibratome (VT1200, Leica Biosystems, Germany) and incubated in preparation medium for 10 min before plating. Three to four sections were gathered per insert, the transfer of which was facilitated by a fire-polished wide mouth glass pipette. Tissue sections were cultured in a growth medium containing Neurobasal (L-Glutamine) supplemented with 2% serum-free B-27, 2% Anti-Anti, 10 mM Glucose, 1 mM MgSO$_4$, and 1 mM Glutamax, at 5% CO$_2$ and 37 °C. Culture medium was replaced 24 h post-plating, and every 48 h thereafter.

**Chemical depletion of microglia from tissue cultures**. Selective depletion of the myeloid cell compartment of human neocortical sections was performed by

supplementing the growth medium with 50 µM of Clodronate (D4434, Milli-poreSigma, St. Louis, MO, USA) for 24 h and 11 µM for the next 48 h.

**Tumor/T-cell inoculation into tissue cultures**. *ZsGreen* tagged BTSC#233 cell lines were cultured as previously described by us[37,72]. Post trypsinization, a centrifugation step was performed, following which the cells were harvested and suspended in MEM media at a concentration of 20,000 cells/µl. inoculate Cultured tissue sections were inoculated with 1 µl of tumor cells using a 10 µl Hamilton syringe, and further cultured as required. Matched peripheral blood samples were collected from the cortical tissue donors. Peripheral T-cells were isolated using the same MACSxpress® Whole Blood Pan T-Cell Isolation Kit (Miltenyi Biotech, Bergisch-Gladbach, Germany) and erythrocytes were eliminated from the suspension using ACK-lysis buffer (Thermo Fisher Scientific, Carlsbad, USA). T-cells were then fluorescently tagged using the CellTrace™ Far Red Cell Proliferation kit (C34564, Thermo Fisher Scientific, Carlsbad, USA) prior to inoculation. Endogenous IL-10 receptor expressed on T-cells were blocked using anti-IL10 neutralizing antibody (ab34843, Abcam, Cambridge, UK) at a concentration of 5 µg/ml for 1 h at 37 °C. Both naïve and neutralized T-cells (40,000 cells/µl) were inoculated into tissue sections and cultured for 48 h.

**Cytokine quantification**. A multi-analyte enzyme-linked immunosorbent assay (ELISA; MEH-003A, Qiagen, Venlo, Netherlands) was used to measure cytokine concentrations of IL2, IL-10, IL-13, and IFN-gamma in the cell culture medium 48 h after T-cell inoculation. Absorbance was determined using a multimode plate reader (Tecan Infinite® 200, Tecan, Männedorf, Switzerland).

**Treatment of patients with JAK-inhibitor**. A patient with a recurrent glioblastoma was treated with a daily dose of 40 mg Ruxolitinib for 4 weeks as neoadjuvant therapy. Pretreatment, tumor progress was confirmed by biopsy. After 4 weeks, the patient underwent a resective surgery along with adjuvant Temozolomide therapy.

**Reporting Summary**. Further information on research design is available in the Nature Research Reporting Summary linked to this article.

## Data availability
The raw data and processed scRNA-seq, stRNA-seq, and RNA-seq data generated in this study have been deposited in the OSF database under accession code https://osf.io/4q32e/ https://doi.org/10.17605/OSF.IO/4Q32E, Source data are available in the source data file. Source data are provided with this paper.

## Code availability
For bulk RNA-seq analysis: https://github.com/heilandd/Vis_Lab1.5, NFCN2: www.github.com/heilandd/NFCN2, SPATA2: https://github.com/theMILOlab/SPATA2. Further information and requests for resources, raw data and reagents should be directed and will be fulfilled by the contact: D.H.Heiland, dieter.henrik.heiland@uniklinik-freiburg.de.

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

## Acknowledgements
D.H.H. is funded by the Else Kröner-Fresenius Foundation. The work is part of the MEPHISTO project (PI: D.H.H. and D.D.), funded by BMBF (German Ministry of Education and Research) (project number: 031L0260B). D.H.H. is funded by the DKTK partner side Freiburg. N.N. was funded by the Müller-Fahnenberg-Stiftung. V.M.R. and K.J. was partially funded by the BMBF-project FMT (13GW0230A). We thank Manching Ku and Dietmar Pfeifer for their helpful technical advice. Part of the illustrations were performed with the help of Biorender.com (Figs. 1, 4, 5, 6 and Supplementary Fig. 7), a paid license of Biorender is available.

## Author contributions
The study was designed and coordinated by D.H.H. Single-cell experiments were performed by P.W., N.N. and V.M.R, Spatial transcriptomics was performed by V.M.R. and P.W. Software development NFCN2: D.H.H., Software development and analysis spatial and single-cell analysis D.H.H., J.Kuc., J.Ker. and D.D. Manuscript writing D.H.H. Slice culture experiments and imaging was performed by V.M.R. and K.J. Editing of the manuscript was performed by D.H.H., V.M.R., O.S., J.B., B.B., T.W., P.F., N.S., U.G.H. and C.F. IMC was performed by B.B. and H.S. IMC analysis and data integration: D.H.H., Neuropathological analysis and IHC was performed by M.P. and R.S., Clinical treatment and follow up C.D. and F.S., FACS and Cell Sorting: M.F., P.W., L.V. and J.P.M., Cell Culture J.M.G., N.N. S.P.B. and M.S.C., RNA sequencing and analysis N.N. and D.H.H.

## Funding

## Competing interests
D.H.H. received reimbursement of travel expenses from 10X and the MILO laboratory is part of the 10X VEP program. The authors declare no competing interests.

## Additional information

[1]Microenvironment and Immunology Research Laboratory, Medical Center-University of Freiburg, Freiburg, Germany. [2]Department of Neurosurgery, Medical Center-University of Freiburg, Freiburg, Germany. [3]Faculty of Medicine, University of Freiburg, Freiburg, Germany. [4]Neuroelectronic Systems, Medical Center-University of Freiburg, Freiburg, Germany. [5]Freiburg Institute for Advanced Studies (FRIAS), University of Freiburg, Freiburg, Germany. [6]Translational NeuroOncology Research Group, Medical Center-University of Freiburg,

Freiburg, Germany. [7]Department of Medicine I, Medical Center-University of Freiburg, Freiburg, Germany. [8]Institute of Medical Bioinformatics and Systems Medicine, Medical Center-University of Freiburg, Freiburg, Germany. [9]Comprehensive Cancer Center Freiburg (CCCF), Faculty of Medicine and Medical Center-University of Freiburg, Freiburg, Germany. [10]German Cancer Consortium (DKTK), partner site Freiburg, Freiburg, Germany. [11]Department of Neurology and Brain Tumor Center, University Hospital Zurich and University of Zurich, Zurich, Switzerland. [12]Department of Neurosurgery, RWTH University of Aachen, Aachen, Germany. [13]Neurosurgical Artificial Intelligence Laboratory Aachen (NAILA), Department of Neurosurgery, RWTH University of Aachen, Aachen, Germany. [14]Department of Anesthesiology and Critical Care Medicine, Medical Center-University of Freiburg, Freiburg, Germany. [15]Institute of Neuropathology, Medical Center-University of Freiburg, Freiburg, Germany. [16]Signalling Research Centre BIOSS and CIBSS, University of Freiburg, Freiburg, Germany. [17]Center for NeuroModulation (NeuroModul), University of Freiburg, Freiburg, Germany. [18]Department of Medicine II: Gastroenterology, Hepatology, Endocrinology, and Infectious Disease, Medical Center- University of Freiburg, Freiburg, Germany. [19]These authors contributed equally: Vidhya M. Ravi, Nicolas Neidert, Paulina Will, Oliver Schnell, Dieter Henrik Heiland. ✉email: dieter.henrik.heiland@uniklinik-freiburg.de

