## [Peer Review File · Nature Communications]

T-Cell Dysfunction in the Glioblastoma Microenvironment is Mediated by Myeloid Cells Releasing Interleukin-10REVIEWER COMMENTS

Reviewer #1 (Remarks to the Author):

This manuscript studies the crosstalk between infiltrating microglia/macrophages and T-cells in the immunosuppressive microenvironment of glioblastoma. This is an important topic that can offer further insights into the mechanisms of resistance to anti-PD1 immunotherapy (Zhao et al. *Nat. Medicine* 25 (2019)). This work is a continuation of previously published work (Herik Heiland et al., *Nat. Commun.* 10 (2019)), where the authors found that the crosstalk between microglia cells and reactive astrocytes is responsible for upregulating IL-10 release in glioblastoma through JAK/STAT signaling. In this manuscript, the authors show that IL-10 secreted by HMOX1+ myeloid cells is responsible for inducing a dysfunctional state in T cells infiltrating mesenchymal regions of the tumor.

There are several aspects that in my opinion need to be addressed before publication:

1. Some of the statements in the Abstract, Introduction, and Discussion do not seem to have clear support in the data and analyses that are presented in the manuscript. Either those statements need to be more carefully crafted or the data and analyses supporting them need to be more clearly presented:

1a. Lines 375-378: the authors state that their analysis indicates that the dysfunctional state of T cells appears to be a transient state. However, this reviewer was unable to find any data or analyses supporting the transitory character of dysfunctional T cell states in subsection "Dysfunctional State of T cells is Driven by IL-10 Signaling".

1b. Lines 58, 117-119, and 394-396: the authors mention that HMOX1+ myeloid cells co-localize spatially with the mesenchymal signature of glioblastoma. However, Figs. 3f,g only show the spot-wise correlation between dysfunctional T cell gene expression markers (HAVCR2 and LAG3) and the mesenchymal gene expression signature, but as far as I see they do not present any analysis of the spot-wise correlation between HMOX1+ myeloid cell markers and the mesenchymal gene expression signature. Similarly, in subsection "T cell Activation and Exhaustion Reveals Spatial Heterogeneity and Association with Glioblastoma Subtypes" there seem to be no results about the spatial pattern of HMOX1+ myeloid cells.

1c. Lines 403-405. The authors state that their experiments with the ex vivo neocortical GBM model confirm that HMOX1+ myeloid cells cause a reduction of effector T cells. However, in these experiments both HMOX1+ and HMOX1- myeloid cells are depleted using clodronate. In absence of other data, these experiments only show that myeloid cells cause a reduction of effector T cells.

1d. Lines 352-356 and 414-418, and Fig 4q-r. It is unclear what comparison was performed here. How did the authors determine that there is a significant enrichment for activated T cells and B cells upon JAK/STAT inhibition in the patient? A more detailed explanation of the comparison and the assumptions that were made would be useful here.

2. The authors introduce the "nearest functionally connected neighbor" algorithm to infer candidate paracrine interactions from the single-cell RNA-seq data. Since the performance of this algorithm has not been rigorously evaluated, it is hard to know how reliable the results of this method are in general. In Supplementary Fig. 5, the authors show that the more conventional and established algorithm NicheNet (Browaeys et al. *Nat. Methods* 17 (2020)) also finds the candidate interaction between myeloid cells (expressing IL10) and T-cells (expressing IL10RA). However, NicheNet does not show that this interaction is specific to HMOX1+ myeloid cells. Can the authors present any other evidence from the single-cell RNA-seq data to support their statement that HMOX1+ cells are mostly responsible for this interaction (e.g. correlation between HMOX1+ and IL10 expression within myeloid cells)? It would be also useful to show the UMAP representation labeled by HMOX1 expression.

3. The description of the "nearest functionally connected neighbor" algorithm in the Methods section

lacks much technical detail. It would be useful to include details about the models, fitting methods, etc., and rewrite the description of the algorithm more carefully.

4. I found the main figures to be unnecessarily complex with 10-18 panels each. The authors might consider keeping the panels that convey the main results and moving the rest of the panels to supplementary figures. There are also several typos that need to be corrected. For example, the panels in Fig. 3 are mismatched with the figure legend (e.g. 3c and 3d seem to be exchanged) and with the main text (lines 283-310). Supplementary Fig. 6 includes a caption "Supplementary Fig. 5" which should be removed. The legend of Supplementary Fig. 3 says "Dimensional reduction (UMAP) of gene expression of the different simulation experiments". However, what is shown in the figure seems to be the UMAP of the single-cell RNA-seq data from the patients colored by the imputed stimulation signatures.

Reviewer #2 (Remarks to the Author):

NCOMMS-21-07876-T

Crosstalk between lymphoid and myeloid cells orchestrates glioblastoma immunity through Interleukin 10 signaling

In their manuscript titled "Crosstalk between lymphoid and myeloid cells orchestrates glioblastoma immunity through Interleukin 10 signaling", investigators leveraged single-cell and spatial transcriptomics to infer cellular crosstalk between macrophages/microglia and T cells within human GBM samples. From eight GBM samples, 21 clusters were identified, where many of these clusters represented macrophages and microglia. Sub-clustering of the T cell cluster revealed different activation states that were then mapped with pseudo-time and RNA velocity analyses. This provided a differentiation map from naïve to terminally exhausted, with an intermediate highly proliferative state. Additionally, by combining these techniques, authors identified an association between the mesenchymal GBM subtype and T cell exhaustion. Furthermore, authors developed a new model, nearest functional connected neighbor, to identify ligand/receptor interactions from scRNAseq data. Computational analyses are thorough and novel; however, the follow-up mechanistic studies, using an in vitro neocortical model, were limited in their support of computational findings. Primary concerns are related to the impact of the mechanistic studies. These concerns are detailed below:

1. A major limitation is the in vitro neocortical model used in Figure 4. Authors state that this model replicates the tumor microenvironment because it is derived from brain tissue; however, during tumorigenesis the TME is largely shaped by the infiltrating immune cells (1), which are absent in this model. Authors should comment on the "myeloid" cells that exist within the neocortical tissue that are being evaluated in figure 4, which should primarily be microglia and not the HMOX1+ macrophages from the computational studies. Along these lines, a quantification of tumor growth in the myeloid depleted condition is important, as myeloid cells generally promote tumor growth, so their absence alone may have an effect on tumor growth independent of T cells or IL10 inhibition.

2. Additionally, the short incubation time (3 days) from tumor injection and subsequent T cell transfer may not provide sufficient time for meaningful interactions and subsequent functional outputs to occur, for example T cell exhaustion. Regarding the transferred T cells, it was not clearly stated whether there is selection for tumor-specific T cells from patient blood, therefore T cells that are injected into the neocortical model may or may not react to the tumor. Evidence of T cell recognition of tumor cells through killing or activation is necessary to increase the impact of this model. Much of T cell exhaustion biology is ignored in this model, such as the conditions and locations under which priming occur and trafficking to the tumor site. Therefore, conclusions that can be drawn from this model are limited. Furthermore, authors should address potential allogeneic reactivity to the cell line BTSC#233 by patient T cells.

3. Although important for spatial information, the immunofluorescence images shown without

quantifications in figure 4 are not sufficient to validate the computational studies. For example, in figure 4g, TIM3 is used to identify “exhausted T cells”, but it has been shown that TIM3 can also be a marker of terminal effector differentiation. Therefore, this would be more convincing if other parameters were used to identify this population, such as PD1 or other functional studies showing T cell activity.

4. When looking at the contribution of different patients to the final clusters, it is apparent that many clusters are specific to single patients. In particular, 80% of the HMOX1+ group is made up of a single patient. What impact does this have on the broader applicability of these findings? Would this bias for a single patient carry over into the downstream analyses? Is this to be expected whenever pooling human samples?

5. The approval for the use of an un-licensed drug in a GBM patient needs to be specifically addressed within the ethics section.

6. Minor concerns:

a. The conclusion that myeloid and lymphoid interactions lead to T cell dysfunction through IL10 is vague and not unoriginal. Secretion of IL10 by myeloid cells is not a novel finding, nor is the role of IL10 in T cell dysfunction(2,3).

b. Using in vitro stimulated T cells to compare with in vivo T cells coming from a tumor ignores much of the complexity in signals that is occurring intratumorally.

c. Using a second cell line in the in vitro neocortical studies would increase the impact of related findings.

In general, authors show novel and interesting computational analyses from cutting edge techniques, however they lack substance in their follow up mechanistic studies. The use of pseudotime and RNA velocity to interrogate T cell activation states and pair them with GBM subtypes and spatial information are intriguing. Additionally, the nearest functionally connected neighbor algorithm will add to the expanding pool of resources for inferring intercellular communication from scRNAseq data. Follow-up studies are limited in their support of computational findings (IL-10 mediated T cell exhaustion); therefore, additional validation studies are required, or the focus of the story should shift to highlight the novel bioinformatic analyses.

Reference:

1. Salmon, H., Remark, R., Gnjatic, S. and Merad, M., 2019. Host tissue determinants of tumour immunity. *Nature Reviews Cancer*, 19(4), pp.215-227.
2. McLane, L.M., Abdel-Hakeem, M.S. and Wherry, E.J., 2019. CD8 T cell exhaustion during chronic viral infection and cancer. *Annual review of immunology*, 37, pp.457-495.
3. Quail, D.F. and Joyce, J.A., 2017. The microenvironmental landscape of brain tumors. *Cancer cell*, 31(3), pp.326-341.

Reviewer #3 (Remarks to the Author):

Ravi, Neidert, Will et al present a single-cell RNA-sequencing study of tumor-infiltrating lymphocytes of patients with GBM. The profile 8 patients and additionally show data for 3 additional patients using spatial profiling, all using 10x platforms. Using established and novel analytical tools to infer trajectories of cell differentiation, they identify variability among T cells; among other, they find sub-clusters of CD8+ T cells with high expression of dysfunction marker TIM3 (HAVCR2) and cells with a hypoxia signature with distinct trajectories. They correlate these signatures with signatures of T cells collected following in vitro stimulation with different cytokines, including IL2, IFNG and IL10, arguing that IL10 stimulated cells have a lower activation score compared to other cells identified as effector cells. Using spatial RNA-seq of three tumors, they find an association of tumor-mesenchymal niches

and infiltration of exhausted T cells (TIM3/LAG3), suggesting that niches of dysfunctional T cells may be in part explained by cancer cell intrinsic features. In order to understand the origin of IL10 and to solidify the role of T cells as IL10 recipients, they present an analytical framework that infers ligand-receptor interactions using several constraints, and demonstrate that myeloid cells (CD163+, HMOX1+) are a main source of IL10. To begin validating this finding, they use slice cultures that they deplete of myeloid cells and show that depletion of myeloid cells results in reduction of IL10 (in the presence of tumor cells); in these models, they then co-culture autologous T cells and show that depletion of myeloid cells in slice cultures results in increased IL2, but not IFNG protein abundance. Incubation of T cells with an IL10R-inhibitor prior to co-culture with tissues results in increased IL2 production in T cells. Because the JAK/STAT pathway is downstream of IL10 signaling, they use ruxolitinib (a selective JAK1/2 inhibitor) first in a slice model showing increased IL2, and then use this drug in the neo-adjuvant therapy of a patient with GBM, followed by analysis of the surgical specimen, which shows activation of T cells.

GBM is a disease with extremely poor prognosis, and therapeutic development has in part been hampered by limited understanding of the tumor microenvironment; as such, the study of potentially high importance. However, several aspects raised significant concerns and reduced enthusiasm for this study, and need to be addressed,

Major points:

1. Nowhere in the manuscript do the authors describe the characteristics of the patient tumors used for either single-cell sequencing or spatial sequencing. Are these all treatment naive tumors? where they exposed to different therapies (radiation, chemotherapy, immunotherapy, investigational drugs) - this will have a dramatic impact on the measured T cell phenotypes and in and of itself could describe variability seen in the data set. The authors should describe basic demographics and treatment history; it is not reasonable to request from the authors to attempt to account for variability based on basic demographics (this, and virtually any single-cell study would be underpowered), but they should show major analyses/findings in the context of different therapies received to exclude the possibility noted above.
2. Technical quality: it is somewhat surprising that the authors only recover ~1000 unique genes per cell with only ~2300 unique molecular identifiers - this is not on par with the quality described and raises concerns regarding data quality; this is particularly surprising as they used the 3.1 chemistry which performs better than prior chemistries; in fact recent studies performing profiling from frozen tissues even achieved similar or better quality compared to this present study (Slyper, Nature Medicine, 2020). Furthermore, some of the clusters described might be artifactual due to tissue processing (e.g. "hypoxia cluster")- this possibility should be addressed using available data sets systematically investigating such artifacts (e.g. Ido Amit laboratory). Furthermore, the authors should comment on the technical quality. An additional embedding showing the UMI count for major analyses should be shown in the supplement to exclude the possibility of technical artifacts as drivers of clustering.
3. There is no statistical evaluation of the inference made in Figure 3f - the authors should provide this; in this same figure, they also show that CCL2 myeloid cells are scoring highly, which is a gene considered to be an immunostimulatory gene/protein - how do they reconcile this? This brings up the question about a more nuanced annotation of the myeloid cells beyond monocytes, macrophages and microglia. The effect size of the gene set enrichment analysis in 3h is very underwhelming. In fact, throughout this section and the studies shown in Figure 4, the effect sizes are very small with in part borderline significance, raising the question of biological significance of these findings.
4. The experiments in slice cultures should be described in more detail in the main text. Here, they state that they performed "myeloid depletion" when in fact they performed microglia depletion (as stated as header in the methods section). The effect size of myeloid/microglia depletion on IL10 production is rather modest. One missing control is depletion of other cell types within the slice culture and measurement of the effect on IL10 to exclude the possibility that there are other major sources of

IL10 production (which is likely). Furthermore, it is surprising that IL2 production, but not IFNG production increases during T cell depletion - the authors should offer potential explanations as this argues against reinvigoration of T cell poly-functionality. The results shown in 4j confirm that IL10 is an immunosuppressive cytokine, but do not substantiate claims that this is mediated by myeloid cells. Again, perplexing that no change in interferon gamma is seen. The single patient study is encouraging - was this pre-/post-comparison performed after single-agent therapy with ruxolitinib or was there a combination used? If the latter, it is possible that observed effects are due to other treatment constituents (see comment 1).

Point – by - Point

D. H. Heiland

We thank all reviewers for their time and effort in evaluating our manuscript and appreciate the positive feedback on our project. We have tried to mitigate the issues highlighted by the reviewers, which has led to a significant improvement in the quality of our manuscript. The following main changes have been made in this context:

1. Quality of the scRNA-seq experiments.

By resequencing the libraries, the quality of the entire dataset was significantly improved and the number of detected genes as well as the UMIs per cell were significantly increased.

2. Analysis and data integration

By applying more advanced algorithms for horizontal and vertical data integration as well as cell type alignment, we were able to present a clearer picture of T cell diversity in the tumors. We separated CD4 and CD8 positive T cells for all downstream analysis.

3. Avoid overfitting in the NFCN algorithm

Our cell communication algorithm has been optimized to reduce potential overfitting and improve prediction. For this purpose, we integrated multiple prediction/validation layers and external algorithms. The new version is also compatible with the conventional scRNA-seq tools (Seurat) and available as an R package (*NFCN2*).

4. Structure of the manuscript and presentation

We restructured our manuscript to present clear hypothesis-driven argumentation and pointed out limitations and ambiguities. The illustrations have been simplified to improve general understanding.

5. Validation model

Our experimental model is not without limitations, which are discussed in detail. We performed new experiments and analysis to improve the experimental validation.

6. Clinical Dataset

Our in-vivo dataset is now described in detail, and we were able to generate further data to strengthen our hypothesis.

In order to discuss the reviewer comments in detail, we provide a point-by-point discussion.

Reviewer #1 (Remarks to the Author):

This manuscript studies the crosstalk between infiltrating microglia/macrophages and T-cells in the immunosuppressive microenvironment of glioblastoma. This is an important topic that can offer further insights into the mechanisms of resistance to anti-PD1 immunotherapy (Zhao et al. Nat. Medicine 25 (2019)). This work is a continuation of previously published work (Henrik Heiland et al., Nat. Commun. 10 (2019)), where the authors found that the crosstalk between microglia cells and reactive astrocytes is responsible for upregulating IL-10 release in glioblastoma through JAK/STAT signaling. In this manuscript, the authors show that IL-10 secreted by HMOX1+ myeloid cells is responsible for inducing a dysfunctional state in T cells infiltrating mesenchymal regions of the tumor.

We would like to thank the reviewer for his time and comments leading to an improvement of the manuscript.

There are several aspects that in my opinion need to be addressed before publication:

1. Some of the statements in the Abstract, Introduction, and Discussion do not seem to have clear support in the data and analyses that are presented in the manuscript. Either those statements need to be more carefully crafted or the data and analyses supporting them need to be more clearly presented:

We have substantially revised our argumentation to better support the data presented and to clearly define our hypotheses. By improving the scRNA-seq datasets and analysis approaches, some of our previously stated hypotheses have been relativized. The major difference compared to our previous manuscript is a separation of CD8 and CD4 positive T cells for all further sub analysis. Our new data incorporated novel aspects of the underlying mechanism of tumor-associated T cell response.

1a. Lines 375-378: the authors state that their analysis indicates that the dysfunctional state of T cells appears to be a transient state. However, this reviewer was unable to find any data or analyses supporting the transitory character of dysfunctional T cell states in subsection "Dysfunctional State of T cells is Driven by IL-10 Signaling".

In our revised version, we performed model integration of RNA-velocity and lineage tree reconstruction to improve the exploration of state specific pathway activation. We found that IL10 response was highly correlated with the expression of exhaustion programs in two T cell clusters. We rewrote this part of the manuscript to improve understanding and removed statements that are no longer supported or have caused confusion.

1b. Lines 58, 117-119, and 394-396: the authors mention that HMOX1+ myeloid cells co-localize spatially with the mesenchymal signature of glioblastoma. However, Figs. 3f,g only show the spot-wise correlation between dysfunctional T cell gene expression markers (HAVCR2 and LAG3) and the mesenchymal gene expression signature, but as far as I see they do not present any analysis of the

spot-wise correlation between HMOX1+ myeloid cell markers and the mesenchymal gene expression signature. Similarly, in subsection “T cell Activation and Exhaustion Reveals Spatial Heterogeneity and Association with Glioblastoma Subtypes” there seem to be no results about the spatial pattern of HMOX1+ myeloid cells.

We thank the reviewer for picking up on this lack of clarity in presentation. We have improved the text and figure legends for clarity. In our revised version of the manuscript, we added spot-wise correlations to support our hypothesis as well as spatial data analysis of the model.

1c. Lines 403-405. The authors state that their experiments with the ex vivo neocortical GBM model confirm that HMOX1+ myeloid cells cause a reduction of effector T cells. However, in these experiments both HMOX1+ and HMOX1- myeloid cells are depleted using clodronate. In absence of other data, these experiments only show that myeloid cells cause a reduction of effector T cells.

Thank you for this helpful comment. It is indeed the case that we remove all myeloid cells and therefore are unable to differentiate HMOX1 pos/neg myeloid cells individually. Using our human model, we are currently not able to specifically target HMOX1 positive cells. We have described these limitations. To approach this limitation, we quantified the spatial distance of HMOX1-positive and -negative cells in our model and concluded that HMOX1-positive cells are mainly localized in the proximity of the tumor. Thus, we assumed that HMOX1-negative cells were only present to a small extent within the tumor. However, in our more detailed analysis of the slice model, we discuss limitations and cofounders more detailed.

1d. Lines 352-356 and 414-418, and Fig 4q-r. It is unclear what comparison was performed here. How did the authors determine that there is a significant enrichment for activated T cells and B cells upon JAK/STAT inhibition in the patient? A more detailed explanation of the comparison and the assumptions that were made would be useful here.

This part was fully rewritten for an improved presentation of our hypothesis. The data are re-analyzed in accordance with our new findings.

2. The authors introduce the “nearest functionally connected neighbor” algorithm to infer candidate paracrine interactions from the single-cell RNA-seq data. Since the performance of this algorithm has not been rigorously evaluated, it is hard to know how reliable the results of this method are in general. In Supplementary Fig. 5, the authors show that the more conventional and established algorithm NicheNet (Browaeys et al. Nat. Methods 17 (2020)) also finds the candidate interaction between myeloid cells (expressing IL10) and T-cells (expressing IL10RA). However, NicheNet does not show that this interaction is specific to HMOX1+ myeloid cells. Can the authors present any other evidence from the single-cell RNA-seq data to support their statement that HMOX1+ cells are mostly responsible

for this interaction (e.g. correlation between HMOX1+ and IL10 expression within myeloid cells)? It would be also useful to show the UMAP representation labeled by HMOX1 expression.

In the new version of our “nearest functionally connected neighbor” (NFCN) algorithm, we implemented various new functions. In general, NFCN is built to quantify cellular interactions of a defined pathway (in our case the IL10-IL10R interaction). In comparison to NicheNet and CellChat, we inferred cellular connections based on the likelihood of cell pairs from the scRNA-seq dataset. Indeed, this quantification leads to overfitting as long as the ground truth is unknown. In order to overcome this problem, we redesigned the algorithm to integrate 3 data layers for improved prediction of cellular interactions.

1. Prediction of the cell-pair likelihood based on scRNA-seq data.

2. Deconvolution of Cell-Cell signaling from doublets

3. Integration of spatial resolved transcriptomics to confirm spatial juxta positioning of cell pairs.

We further integrated an unsupervised model using CellChat to infer the most common Cell-Cell interaction across clusters. Through our optimization, we tailored the model to predict cellular communication and reduced bias. We have added a supplementary result part to explain this model in detail.

3. The description of the “nearest functionally connected neighbor” algorithm in the Methods section lacks much technical detail. It would be useful to include details about the models, fitting methods, etc., and rewrite the description of the algorithm more carefully.

As mentioned in the answer above, we added supplementary results with detailed information.

4. I found the main figures to be unnecessarily complex with 10-18 panels each. The authors might consider keeping the panels that convey the main results and moving the rest of the panels to supplementary figures. There are also several typos that need to be corrected. For example, the panels in Fig. 3 are mismatched with the figure legend (e.g. 3c and 3d seem to be exchanged) and with the main text (lines 283-310). Supplementary Fig. 6 includes a caption “Supplementary Fig. 5” which should be removed. The legend of Supplementary Fig. 3 says “Dimensional reduction (UMAP) of gene expression of the different simulation experiments”. However, what is shown in the figure seems to be the UMAP of the single-cell RNA-seq data from the patients colored by the imputed stimulation signatures.

Thanks for pointing out the typos and complexity of the figures. We have adapted the figures to facilitate ease of understanding.

Reviewer #2 (Remarks to the Author):

Crosstalk between lymphoid and myeloid cells orchestrates glioblastoma immunity through Interleukin 10 signaling. In their manuscript titled “Crosstalk between lymphoid and myeloid cells orchestrates glioblastoma immunity through Interleukin 10 signaling”, investigators leveraged single-cell and spatial transcriptomics to infer cellular crosstalk between macrophages/microglia and T cells within human GBM samples. From eight GBM samples, 21 clusters were identified, where many of these clusters represented macrophages and microglia. Sub-clustering of the T cell cluster revealed different activation states that were then mapped with pseudo-time and RNA velocity analyses. This provided a differentiation map from naïve to terminally exhausted, with an intermediate highly proliferative state. Additionally, by combining these techniques, authors identified an association between the mesenchymal GBM subtype and T cell exhaustion. Furthermore, authors developed a new model, nearest functional connected neighbor, to identify ligand/receptor interactions from scRNAseq data. Computational analyses are thorough and novel; however, the follow-up mechanistic studies, using an in vitro neocortical model, were limited in their support of computational findings. Primary concerns are related to the impact of the mechanistic studies. These concerns are detailed below:

We would like to thank the reviewer for his time and comments leading to an improvement of the manuscript.

1. A major limitation is the in vitro neocortical model used in Figure 4. Authors state that this model replicates the tumor microenvironment because it is derived from brain tissue; however, during tumorigenesis the TME is largely shaped by the infiltrating immune cells (1), which are absent in this model. Authors should comment on the “myeloid” cells that exist within the neocortical tissue that are being evaluated in figure 4, which should primarily be microglia and not the HMOX1+ macrophages from the computational studies. Along these lines, a quantification of tumor growth in the myeloid depleted condition is important, as myeloid cells generally promote tumor growth, so their absence alone may have an effect on tumor growth independent of T cells or IL10 inhibition.

Thank you for this valuable comment. We have addressed and discussed this limitation in detail. There is no doubt that the myeloid cells within the presented model are predominantly composed of microglial cells. However, these cells can also transform reactively and consequently become HMOX1 positive. HMOX1 positive microglial cells also play a crucial role in other pathologies such as traumatic brain injury and subarachnoid hemorrhage. Therefore, it would be safe to assume that although the model is limited, the specific role associated with HMOX1 expression can be associated with activated microglial cells. We have discussed this limitation in detail.

Regarding the quantification of tumor growth in myeloid depletion condition: This question is of high interest and our laboratory is currently working on this interaction. At the moment, we feel that the addition of this data will result in a loss of focus of the results presented in this manuscript.

2. Additionally, the short incubation time (3 days) from tumor injection and subsequent T cell transfer may not provide sufficient time for meaningful interactions and subsequent functional outputs to occur, for example T cell exhaustion. Regarding the transferred T cells, it was not clearly stated whether there is selection for tumor-specific T cells from patient blood, therefore T cells that are injected into the neocortical model may or may not react to the tumor. Evidence of T cell recognition of tumor cells through killing or activation is necessary to increase the impact of this model. Much of T cell exhaustion biology is ignored in this model, such as the conditions and locations under which priming occur and trafficking to the tumor site. Therefore, conclusions that can be drawn from this model are limited. Furthermore, authors should address potential allogeneic reactivity to the cell line BTSC#233 by patient T cells.

Thank you for the detailed review of the model, which aids illustrating the various aspects, functionalities and limitations. Indeed, we are limited in the interpretation of our results. However, the following points deserve to be considered:

1. Regarding the first part of the question: We did not isolate tumor-specific T cells (mutation-associated neoantigens (MANA) associated TILs) from blood. In the context of brain tumors, to purify MANA-TILs is extremely challenging and only insufficiently possible using current methods. The aim of our model was to generate a T cell response and investigate the role of the tumor-associated microenvironment. Injection of a primary cell line which causes an allogeneic response is part of the model. Without this stimulus, a T cell response, as you mentioned above, is limited. This allogeneic reactivity should therefore be considered as intentional.

2) Regarding the second part of the question: Our data show that T cell activity (GZMB) in the tumor region becomes detectable after 3 days. (See data presented). The temporal dimensions of our slice model span a few days because the tumor infiltrates a large portion of the slice within 7 days. We have already reported tumor growth times in our previous publications^{1,2}.

3. Although important for spatial information, the immunofluorescence images shown without quantifications in figure 4 are not sufficient to validate the computational studies. For example, in figure 4g, TIM3 is used to identify “exhausted T cells”, but it has been shown that TIM3 can also be a marker of terminal effector differentiation. Therefore, this would be more convincing if other parameters were used to identify this population, such as PD1 or other functional studies showing T cell activity.

We added a more sophisticated validation of the imaging results. We choose Tim3 to confirm the results from the computational studies, in which the tissue resident memory cluster revealed the strongest enrichment of exhaustion markers.

4. When looking at the contribution of different patients to the final clusters, it is apparent that many clusters are specific to single patients. In particular, 80% of the HMOX1+ group is made up of a single patient. What impact does this have on the broader applicability of these findings? Would this bias for a single patient carry over into the downstream analyses? Is this to be expected whenever pooling human samples?

This problem was based on the vertical integration algorithm which has been fully revised.

5. The approval for the use of an un-licensed drug in a GBM patient needs to be specifically addressed within the ethics section.

The treatment was performed as part of the "Compassionate Use" program (RL 2001/83/EG VO 726/2004). We added explanations in the manuscript.

6. Minor concerns: a. The conclusion that myeloid and lymphoid interactions lead to T cell dysfunction through IL10 is vague and not unoriginal. Secretion of IL10 by myeloid cells is not a novel finding, nor is the role of IL10 in T cell dysfunction(2,3).

Indeed, this mechanism is reported in other cancer types but not for brain malignancy so far. Other cancer types can also be treated with checkpoint inhibitors, which is not possible for GBM. We think that investigating this special environment expands our comprehension. The fact that we found similar mechanism that can be also observed in other cancer types is not unexpected.

b. Using in vitro stimulated T cells to compare with in vivo T cells coming from a tumor ignores much of the complexity in signals that is occurring intratumorally.

The stimulation experiments are used to detect downstream pathway activation based on an isolated cytokine. We remove all other interpretations.

c. Using a second cell line in the in vitro neocortical studies would increase the impact of related findings.

Since this work does not focus on the tumor directly, using multiple donors to investigate the variance across patients was our main focus.

In general, authors show novel and interesting computational analyses from cutting edge techniques, however they lack substance in their follow up mechanistic studies. The use of pseudotime and RNA velocity to interrogate T cell activation states and pair them with GBM subtypes and spatial information are intriguing. Additionally, the nearest functionally connected neighbor algorithm will add to the expanding pool of resources for inferring intercellular communication from scRNAseq data. Follow-up

studies are limited in their support of computational findings (IL-10 mediated T cell exhaustion); therefore, additional validation studies are required, or the focus of the story should shift to highlight the novel bioinformatic analyses.

Thank you for the appreciation of the computational results and tools. However, we think that biological validation, even if limited, supports the computational analysis.

Reviewer #3 (Remarks to the Author):

Ravi, Neidert, Will et al present a single-cell RNA-sequencing study of tumor-infiltrating lymphocytes of patients with GBM. The profile 8 patients and additionally show data for 3 additional patients using spatial profiling, all using 10x platforms. Using established and novel analytical tools to infer trajectories of cell differentiation, they identify variability among T cells; among other, they find sub-clusters of CD8+ T cells with high expression of dysfunction marker TIM3 (HAVCR2) and cells with a hypoxia signature with distinct trajectories. They correlate these signatures with signatures of T cells collected following in vitro stimulation with different cytokines, including IL2, IFNG and IL10, arguing that IL10 stimulated cells have a lower activation score compared to other cells identified as effector cells. Using spatial RNA-seq of three tumors, they find an association of tumor-mesenchymal niches and infiltration of exhausted T cells (TIM3/LAG3), suggesting that niches of dysfunctional T cells may be in part explained by cancer cell intrinsic features. In order to understand the origin of IL10 and to solidify the role of T cells as IL10 recipients, they present an analytical framework that infers ligand-receptor interactions using several constraints, and demonstrate that myeloid cells (CD163+, HMOX1+) are a main source of IL10. To begin validating this finding, they use slice cultures that they deplete of myeloid cells and show that depletion of myeloid cells results in reduction of IL10 (in the presence of tumor cells); in these models, they then co-culture autologous T cells and show that depletion of myeloid cells in slice cultures results in increased IL2, but not IFNG protein abundance. Incubation of T cells with an IL10R-inhibitor prior to co-culture with tissues results in increased IL2 production in T cells. Because the JAK/STAT pathway is downstream of IL10 signaling, they use ruxolitinib (a selective JAK1/2 inhibitor) first slice model showing increased IL2, and then use this drug in the neo-adjuvant therapy of a patient with GBM, followed by analysis of the surgical specimen, which shows activation of T cells.

GBM is a disease with extremely poor prognosis, and therapeutic development has in part been hampered by limited understanding of the tumor microenvironment; as such, the study of potentially high importance. However, several aspects raised significant concerns and reduced enthusiasm for this study, and need to be addressed,

We would like to thank the reviewer for his time and comments leading to an improvement of the manuscript.

Major points:

1. Nowhere in the manuscript do the authors describe the characteristics of the patient tumors used for either single-cell sequencing or spatial sequencing. Are these all treatment naive tumors? where they exposed to different therapies (radiation, chemotherapy, immunotherapy, investigational drugs) - this will have a dramatic impact on the measured T cell phenotypes and in and of itself could describe variability seen in the data set.

All samples used for the dataset are naive non-treated primary GBM samples except the JAK-inhibitor treated samples as described in the last section on the manuscript.

The authors should describe basic demographics and treatment history; it is not reasonable to request from the authors to attempt to account for variability based on basic demographics (this, and virtually any single-cell study would be underpowered), but they should show major analyses/findings in the context of different therapies received to exclude the possibility noted above.

Indeed, prior treatment can strongly affect the immune compartment. Here, only primary non-treated samples are included. We have added a supplementary table for demographic details.

2. Technical quality: it is somewhat surprising that the authors only recover ~1000 unique genes per cell with only ~2300 unique molecular identifiers - this is not on par with the quality described and raises concerns regarding data quality; this is particularly surprising as they used the 3.1 chemistry which performs better than prior chemistries; in fact recent studies performing profiling from frozen tissues even achieved similar or better quality compared to this present study (Slyper, Nature Medicine, 2020). Indeed, the sequencing depth of the samples was only moderate (Sequencing Saturation ~10%-15%) and we decided to re-sequence all our libraries. We gained an improvement in quality to approximately ~2400 genes per cell and ~10k unique molecular identifiers. To provide an overview of the dataset quality, we have added some comparisons to recent published datasets, **Supplementary Figure 2**.

Furthermore, some of the clusters described might be artifactual due to tissue processing (e.g. "hypoxia cluster")- this possibility should be addressed using available data sets systematically investigating such artifacts (e.g. Ido Amit laboratory). Furthermore, the authors should comment on the technical quality. An additional embedding showing the UMI count for major analyses should be shown in the supplement to exclude the possibility of technical artifacts as drivers of clustering.

We added a scRNA-seq quality check in the supplementary results. Using a recently reported cell-type alignment algorithm (WNN³), we redesigned the first part and specifically investigated CD8 and CD4 T cells separately. As recommended, we have opted for alignment to reference datasets. The stress cluster within the T cell population has already been confirmed in the meantime in another cohort⁴.

3. There is no statistical evaluation of the inference made in Figure 3f - the authors should provide this; in this same figure, they also show that CCL2 myeloid cells are scoring highly, which is a gene considered to be an immunostimulatory gene/protein - how do they reconcile this? This brings up the question about a more nuanced annotation of the myeloid cells beyond monocytes, macrophages and microglia. The effect size of the gene set enrichment analysis in 3h is very underwhelming. In fact, throughout this section and the studies shown in Figure 4, the effect sizes are very small with in part borderline significance, raising the question of biological significance of these findings.

We have revised large parts of the results presented formerly in Figure 4. This was done by resequencing and improving the vertical integration, have eliminated the previously seen myeloid cell populations. As already mentioned above, the myeloid populations seem to have been an artefact of

insufficient vertical integration. Retrospectively, the integration used at that time by means of the cell ranger pipeline was not a sufficient approach. The MNN integration used now offers a much better approach to data integration.

4. The experiments in slice cultures should be described in more detail in the main text. Here, they state that they performed "myeloid depletion" when in fact they performed microglia depletion (as stated as header in the methods section).

We have tried to discuss the experimental part, especially the limitations, in detail. Our new data and analyses better support the reported results. The chemical depletion of microglia used here, the myeloid population in the human brain slice, has already been described in detail our previous work^{1,2}.

The effect size of myeloid/microglia depletion on IL10 production is rather modest. One missing control is depletion of other cell types within the slice culture and measurement of the effect on IL10 to exclude the possibility that there are other major sources of IL10 production (which is likely).

Indeed, we agree that there are other sources of IL10 release in the tumor microenvironment. Neurons, astrocytes, and oligodendrocytes are potential candidates for IL10 release, but protocols for depletion of these cell types have not yet been established. To address this question, we attempted to establish cell-specific depletion for astrocytes. Unfortunately, we failed with this because the toxicity of astrocyte depletion is too high. Our results with IL10 inhibition confirm the downstream mechanism but cannot conclusively resolve the question of which cell type should also be considered as an IL10 source. At least our data suggest that a large fraction of IL10 is derived from HMOX1-positive myeloid cells.

Furthermore, it is surprising that IL2 production, but not IFNG production increases during T cell depletion - the authors should offer potential explanations as this argues against reinvigoration of T cell poly-functionality.

We agree with the reviewer that the IFNG signal is difficult to explain. When we examined the raw signal, we found generally high levels (including the negative control) of IFNG and other cytokines in the ELISA, suggesting a potential technical problem. We fully reviewed the ELISA data and re-ran the assay. The new data provided a clearer picture. Also in the new data, we see only a small baseline effect on IL10 after depletion of microglia. We assume that within our slice model the reactive transformation of microglia is necessary to upregulate IL10 release. Therefore, differences in the "no-tumor" slices were not observed.

The results shown in 4j confirm that IL10 is an immunosuppressive cytokine, but do not substantiate claims that this is mediated by myeloid cells. Again, perplexing that no change in interferon gamma is seen.

As mentioned above, we are limited within our model to elucidate all potential sources of IL10 release. Our data confirms that a significant part originates from myeloid cells.

The single patient study is encouraging - was this pre-/post-comparison performed after single-agent therapy with ruxolitinib or was there a combination used? If the latter, it is possible that observed effects are due to other treatment constituents (see comment 1).

We used monotherapy in a neoadjuvant setting. However, the patient was pretreated with RT+TMZ +CCNU (CeTeG protocol) and received TTF (not in parallel to ruxolitinib). Resection of the tumor was subsequently performed after 6 weeks of ruxolitinib monotherapy. For analysis, we were able to perform staining for direct pre/post-treatment. Unfortunately, single cell sequencing could not be performed from the very small biopsy sample (before therapy), but only from the surgery sample (after treatment). We added a more detailed description and illustrations in the updated manuscript.

Bibliography

1. Ravi, V. M. *et al.* Human organotypic brain slice culture: a novel framework for environmental research in neuro-oncology. *Life Sci. Alliance* **2**, (2019).
2. Henrik Heiland, D. *et al.* Tumor-associated reactive astrocytes aid the evolution of immunosuppressive environment in glioblastoma. *Nat. Commun.* **10**, 2541 (2019).
3. Hao, Y., Hao, S. & Andersen, E. Integrated analysis of multimodal single-cell data. *Nissen*
4. Mathewson, N. D. *et al.* Inhibitory CD161 receptor identified in glioma-infiltrating T cells by single-cell analysis. *Cell* **184**, 1281–1298.e26 (2021).

REVIEWER COMMENTS

Reviewer #1 (Remarks to the Author):

The authors have substantially improved the presentation of the hypotheses, analyses, and data in the revised manuscript, and have incorporated several new analyses that fill some of the gaps in the previous version. In my opinion, the revised manuscript is suited for publication.

I would only like to point some typos and small suggestions to improve the clarity of some parts:

- Line 123: "reference datasets" -> "reference dataset"
- Line 137: "Supplementary Figure 1b-c" -> "Figure 1b-c"
- Line 189: "Figure 2b" -> "Figure 2c"
- Line 223: "Figure 3d" -> "Figure 3f"
- Line 309: "Figure 5b" -> "Figure 5c"
- Line 389: "Benjamini- Hochberger" -> "Benjamini-Hochberg"
- Line 316: the notation used for the two clusters of myeloid cells (aM\Phi and bM\Phi) does not match with the notation used in Supplementary Fig. 5.
- In figure 1d,e it is unclear what the authors mean by "z-scored Gene Expression". z-scores are not bounded between 0 and 1, so I suspect the authors refer to something else or they have rescaled the z-scores in some way to lie between 0 and 1. I would suggest clarifying the normalization used in the figure legend and in the methods section.
- The specific gene sets from MSigDB v7 that were used should be specified in the methods section.
- I find the notation used in Fig. 5c and other figures (circle color + size) to be confusing. For example, in Fig. 5c the Mes-like correlation of #URK_S3 seems larger than the Mes-like correlation of #UKF_S2 based on the color of the circle, but smaller or equal based on the size of the circle.
- It would be helpful to add more details in the legend of Fig. 5b. For example, what does each of the colors in the figure denote?

Reviewer #2 (Remarks to the Author):

In their revised manuscript titled "Crosstalk between lymphoid and myeloid cells orchestrates glioblastoma immunity through Interleukin 10 signaling", investigators coupled scRNAseq and stRNAseq to query the tumor microenvironment of 8 treatment naïve glioblastoma patients. Ligand/receptor interactions were identified using a novel algorithm, nearest functionally connected neighbors (NFCN), ultimately identifying HMOX1+ myeloid cells as a major source of IL-10. A T cell exhaustion phenotype was linked to the HMOX1+ myeloid cells and validated with both stRNAseq and an ex vivo neocortical system. In this revision, authors addressed concerns regarding the limitations of ex vivo experiments in validation of in silico findings. Additionally, authors increased the detail in many of the results sections to clarify relevant findings. These revisions greatly improve the quality and impact of the manuscript.

Reviewer #3 (Remarks to the Author):

The authors have done a very good job revising the manuscript and addressing my comments and suggestions.

I would strongly encourage them to highlight some of the technical and experimental challenges they had in their revision, as this might be of importance for future studies by their and other groups.

Benjamin Izar

Point – by – Point Revision 2

D. H. Heiland

We thank all reviewers for their time and effort in evaluating our manuscript and appreciate the positive feedback on our project.

Reviewer #1 (Remarks to the Author):

The authors have substantially improved the presentation of the hypotheses, analyses, and data in the revised manuscript, and have incorporated several new analyses that fill some of the gaps in the previous version. In my opinion, the revised manuscript is suited for publication.

I would only like to point some typos and small suggestions to improve the clarity of some parts:

- Line 123: “reference datasets” -> “reference dataset”

Changed

- Line 137: “Supplementary Figure 1b-c” -> “Figure 1b-c”

Changed

- Line 189: “Figure 2b” -> “Figure 2c”

Changed

- Line 223: “Figure 3d” -> “Figure 3f”

Changed

- Line 309: “Figure 5b” -> “Figure 5c”

Changed

- Line 389: “Benjamini- Hochberger” -> “Benjamini-Hochberg”

Changed

- Line 316: the notation used for the two clusters of myeloid cells (aM\Phi and bM\Phi) does not match with the notation used in Supplementary Fig. 5.

Changed in the new supplementary file

- In figure 1d,e it is unclear what the authors mean by “z-scored Gene Expression”. z-scores are not bounded between 0 and 1, so I suspect the authors refer to something else or they have rescaled the z-scores in some way to lie between 0 and 1. I would suggest clarifying the normalization used in the figure legend and in the methods section.

It is normalized gene expression, we changed the figure description

- The specific gene sets from MSigDB v7 that were used should be specified in the methods section.

The gene sets are implemented in the method part.

- I find the notation used in Fig. 5c and other figures (circle color + size) to be confusing. For example, in Fig. 5c the Mes-like correlation of #URK_S3 seems larger than the Mes-like correlation of #UKF_S2 based on the color of the circle, but smaller or equal based on the size of the circle.

It was based on the fact that plots were created individually, we changed this.

- It would be helpful to add more details in the legend of Fig. 5b. For example, what does each of the colors in the figure denotes?

We added a description in the figure.

Reviewer #2 (Remarks to the Author):

In their revised manuscript titled “Crosstalk between lymphoid and myeloid cells orchestrates glioblastoma immunity through Interleukin 10 signaling”, investigators coupled scRNAseq and stRNAseq to query the tumor microenvironment of 8 treatment naïve glioblastoma patients. Ligand/receptor interactions were identified using a novel algorithm, nearest functionally connected neighbors (NFCN), ultimately identifying HMOX1+ myeloid cells as a major source of IL-10. A T cell exhaustion phenotype was linked to the HMOX1+ myeloid cells and validated with both stRNAseq and an ex vivo neocortical system. In this revision, authors addressed concerns regarding the limitations of ex vivo experiments in validation of in silico findings. Additionally, authors increased the detail in many of the results sections to clarify relevant findings. These revisions greatly improve the quality and impact of the manuscript.

Reviewer #3 (Remarks to the Author):

The authors have done a very good job revising the manuscript and addressing my comments and suggestions. I would strongly encourage them to highlight some of the technical and experimental challenges they had in their revision, as this might be of importance for future studies by their and other groups.

R2 & R3: Thank you for the positive response. We are pleased that the reviewers appreciated the considerable effort and the change during the revision process. In our opinion, a description of these changes in the discussion clearly hampers the red-thread and takes the focus away from the scientific results. In our opinion, the detailed process can be reconstructed very well from published reviewers' comments, so we have refrained from making a statement in the discussion part.